# Activation of PTHrP-cAMP-CREB1 signaling following p53 loss is essential for osteosarcoma initiation and maintenance

Mannu K Walia[1], Patricia MW Ho[1], Scott Taylor[1], Alvin JM Ng[1,2], Ankita Gupte[1], Alistair M Chalk[1,2], Andrew CW Zannettino[3,4], T John Martin[1,2*†], Carl R Walkley[1,2,5*†]

[1]St. Vincent's Institute of Medical Research, Fitzroy, Australia; [2]Department of Medicine, St Vincent's Hospital, University of Melbourne, Fitzroy, Australia; [3]Myeloma Research Laboratory, School of Medicine, Faculty of Health Sciences, University of Adelaide, Adelaide, Australia; [4]Cancer Theme, South Australian Health and Medical Research Institute, Adelaide, Australia; [5]ACRF Rational Drug Discovery Centre, St. Vincent's Institute of Medical Research, Fitzroy, Australia

*For correspondence: jmartin@
svi.edu.au (TJM); cwalkley@svi.
edu.au (CRW)

†These authors contributed
equally to this work

Competing interests: The
authors declare that no
competing interests exist.

Reviewing editor: Jonathan A
Cooper, Fred Hutchinson Cancer
Research Center, United States

**Abstract** Mutations in the P53 pathway are a hallmark of human cancer. The identification of pathways upon which p53-deficient cells depend could reveal therapeutic targets that may spare normal cells with intact p53. In contrast to P53 point mutations in other cancer, complete loss of P53 is a frequent event in osteosarcoma (OS), the most common cancer of bone. The consequences of p53 loss for osteoblastic cells and OS development are poorly understood. Here we use murine OS models to demonstrate that elevated *Pthlh (Pthrp)*, cAMP levels and signalling via CREB1 are characteristic of both p53-deficient osteoblasts and OS. Normal osteoblasts survive depletion of both PTHrP and CREB1. In contrast, p53-deficient osteoblasts and OS depend upon continuous activation of this pathway and undergo proliferation arrest and apoptosis in the absence of PTHrP or CREB1. Our results identify the PTHrP-cAMP-CREB1 axis as an attractive pathway for therapeutic inhibition in OS.

## Introduction

Mutations within the P53 pathway occur in all human cancers (*Hanahan and Weinberg, 2011*). While the mutation of *TP53* itself is a common event, how this contributes to the initiation of cancer is incompletely understood. The most prevalent mutations are point mutations that result in proteins with altered function (*Olivier et al., 2010*). Extensive analysis of these mutations using mouse models has revealed the pervasive cellular consequences of mutant P53 (*Bieging and Attardi, 2012; Bieging et al., 2014; Goh et al., 2011*). In osteosarcoma (OS), the most common primary tumour of bone, unique genomic rearrangements and other mutation types most often result in null alleles of P53 (*Ribi et al., 2015; Chen et al., 2014*). The reason for this distinct *TP53* mutational preference in osteoblastic cells, the lineage of origin of OS, is not understood, nor are the signaling cascades that are altered in p53-deficient osteoblastic cells that facilitate the initiation of OS. Understanding how the loss of P53 modifies osteoblast precursor cells to enable OS initiation will provide new avenues to improve clinical outcomes.

OS occurs predominantly in children and teenagers and 5 year survival rates have plateaued at ~70% for patients with localised primary disease and ~20% for patients with metastatic or recurrent

**eLife digest** Bone cancer (osteosarcoma) is caused by mutations in certain genes, which results in cells growing and dividing uncontrollably. In particular, a gene that produces a protein called P53 in humans is lost in all bone cancers. However, we don't understand what happens to the bone cells when they lose P53. Although a number of studies have identified several molecular pathways that are changed in bone cancers – such as the cyclic AMP (cAMP) pathway – how these interact to cause a cancer is not well understood.

Walia et al. compared bone-forming cells from normal mice with cells from mutant mice from which the gene that produces the mouse p53 protein could be removed. This revealed that the loss of p53 causes these cells to grow faster. The activity of the cAMP pathway also increases in p53-deficient cells. Further investigation revealed that the cells grow faster only if they are able to activate the cAMP pathway, and that this pathway needs to stay active for bone cancer cells to grow and survive. This suggests that inhibiting this pathway could present a new way to treat bone cancer.

Walia et al. confirmed several of their findings in human cells. Future studies will now investigate how the loss of the P53 protein in humans activates the cAMP pathway, which will be important for understanding how this cancer forms. It will also be worthwhile to begin testing ways to block this pathway to determine whether it is a useful target for therapies.

disease (*Janeway et al., 2012*; *Mirabello et al., 2009*). The advances in the understanding of OS biology and genetics have brought limited patient benefit to date or changes in clinical management. Sequencing of OS using both whole genome and exome approaches identified the universal mutation of *TP53* accompanied by recurrent mutation of *RB1, ATRX* and *DLG2* in 29%-53% of cases (*Ribi et al., 2015*; *Chen et al., 2014*; *Perry et al., 2014*). The OS predisposition of Li-Fraumeni patients and mouse models support the key role of *p53* mutation in OS: $Trp53^{+/-}$ and $Trp53^{-/-}$ mice develop OS in addition to other tumors while conditional deletion of *Trp53* in the osteoblastic lineage results in full penetrance OS, largely in the absence of other tumor types (*Mutsaers and Walkley, 2014*; *Donehower et al., 1992*; *Quist et al., 2015*; *Wang et al., 2006*; *Lengner et al., 2006*; *Zhao et al., 2015*). The consequence of p53 loss in osteoblastic cells is only understood to a limited extent. A more complete understanding of the pathways impacted by loss of p53 will be important to understanding the rewiring of osteoblastic cells that underlies OS initiation.

Genetic association studies (GWAS) in OS have identified changes in cyclic AMP (cAMP) related processes as predisposing to OS. A GWAS defined two OS susceptibility loci in human: the metabotropic glutamate receptor *GRM4* and a region on chromosome 2p25.2 lacking annotated transcripts (*Savage et al., 2013*). *GRM4* has a role in cAMP generation. A GWAS in dogs with OS identified variants of *GRIK4* and *RANK* (*TNFRSF11A*), both involved in cAMP pathways (*Karlsson et al., 2013*). Using a murine OS model induced by an osteocalcin promoter-driven SV40T/t, OS were identified that deleted a regulatory subunit of the cAMP-dependent protein kinase (PKA) complex, *Prkar1a*, or with amplification of *Prkaca*, the PKA catalytic component (*Molyneux et al., 2010*). A recent transposon mediated mutagenesis OS model recovered both activating and inactivating mutations within cAMP related pathways, but no functional analysis was performed (*Moriarity et al., 2015*). Evidence from multiple species implicates enhanced cAMP-PKA activity in OS. Osteoblastic cells are highly sensitive to cAMP levels, and major regulators of the osteoblast lineage such as PTHrP/PTH increase cAMP and activate cAMP-dependent signaling (*Juppner et al., 1991*). The requirement for these pathways in OS initiation and maintenance has not been tested.

Parathyroid hormone (PTH) and PTH-related protein (PTHrP) are key regulators of osteoblast and skeletal homeostasis (*McCauley and Martin, 2012*; *Martin, 2016*). PTHrP and PTH activate their common receptor, PTHR1 (*Suva et al., 1987*; *Juppner et al., 1988*). Binding to PTHR1 on osteoblasts or OS cells activates adenylyl cyclase, stimulates cAMP production, followed by PKA activation, leading to many of the transcriptional changes associated with PTH/PTHrP treatment (*Gardella and Jüppner, 2001*; *Pioszak and Xu, 2008*; *Swarthout et al., 2002*; *Partridge et al., 1981*). In normal osteoblasts PTHrP is produced by osteoblastic lineage cells and acts in a paracrine

manner upon other osteoblastic cells at different stages of differentiation (*Martin, 2005*; *Miao, 2005*). The long-term administration of PTH(1–34) resulted in a high incidence of OS in rats (*Vahle et al., 2002*). Elevated expression of PTHR1 is a feature of human and rodent OS (*Martin et al., 1976*; *Yang et al., 2007*). PTHrP was expressed in murine OS subtypes (*Ho et al., 2015*), placing it as a plausible ligand for activating PTHR1 signaling. It is presently unknown how PTHrP might act in OS. The consequence of PTH/PTHrP signaling via cAMP-PKA is phosphorylation of the cAMP response element binding (CREB1) protein (*Datta and Abou-Samra, 2009*). CREB1 regulates gene expression through the activation of cAMP-dependent or -independent signal transduction pathways (*Mayr and Montminy, 2001*).

We sought to understand how loss of p53 leads to the initiation of OS. We identified that an early consequence of p53 deletion in osteoblastic cells was increased cAMP levels and the autocrine activation of cAMP signalling via PTHrP. This same signaling node is active in OS and was important in both the initiation and maintenance of OS. The activation of the PTHrP-cAMP-CREB1 axis was required for the hyperproliferative phenotype of *Trp53* deficient osteoblasts and the maintenance of established OS, identifying this as a tractable pathway for therapeutic inhibition in OS.

## Results

### cAMP and CREB1 dependent signaling are activated in *Trp53*-deficient osteoblasts

As inactivating mutations of *TP53* are universal in conventional OS, we used this to model an OS initiating lesion (*Chen et al., 2014*). Primary osteoblasts were isolated from *R26*-CreER$^{T2ki/+}$*Trp53*$^{+/+}$ (WT) and *R26*-CreER$^{T2ki/+}$*Trp53*$^{fl/fl}$ (KO) animals and in vitro tamoxifen treatment was used to induce deletion of p53. Over 20 days culture, a loss of expression of p53 target genes in the KO cultures + tamoxifen occurred, compared to both WT and non-tamoxifen treated isogenic *R26*-CreER$^{T2ki/+}$*Trp53*$^{fl/fl}$ cultures (*Figure 1A*). Given the strong association between osteoblastic differentiation, OS and cAMP signaling, we assessed if pathways were impacted by loss of p53. CREB1 transcriptional target genes were identified from ChIP and ChIP-Chip studies of CREB genomic occupancy (*Kenzelmann Broz et al., 2013*; *Ravnskjaer et al., 2007*). Only those targets that associated with CREB1 in response to cAMP activation were considered. Analogously, p53 target genes were defined from a ChIP-seq dataset from human HCT116 cells (*Sánchez et al., 2014*) and further refined against a second independent dataset of p53 ChIP-seq from murine embryonic fibroblasts (*Kenzelmann Broz et al., 2013*). Strikingly, the expression of CREB1 target genes was increased, inversely paralleling the reduction in p53 target genes (*Figure 1A*, *Figure 1—figure supplement 1A–B*). Similar gene expression results were obtained using shRNA against *Trp53* in primary WT osteoblasts, demonstrating that the observed changes did not result from proliferation differences (*Figure 1—figure supplement 1C–E*). The altered transcript levels were reflected at the protein level, where loss of p53 was associated with an increase in total CREB1 and phosphorylated CREB1 (pCREB1) in the KO cells (*Figure 1B*). Interestingly, the KO cultures had increased cAMP levels compared to WT or isogenic controls (*Figure 1C*). Collectively, these results demonstrate that derepression of cAMP/CREB1 pathways is an early event following *Trp53* mutation in osteoblasts.

The tamoxifen treated *Trp53*-KO osteoblasts hyperproliferate after ∼15 days (*Ng et al., 2015*), coinciding with the loss of p53. Coinciding with the increased proliferation of the p53-deficient cultures was an increase in cAMP per cell and an activation of CREB1 target genes, potentially explained by the increased *Pthlh* expression (also known as *Pthrp*, *Figure 1A–C*). As elevated PTHrP would increase cAMP levels, the involvement of both CREB1 and PTHrP in the p53-deficient response was assessed. Primary osteoblasts from the respective genotypes were infected with two independent shRNAs against either *Creb1* or *Pthlh* then cultured ± tamoxifen. Efficient and stable knockdown of *Creb1* and *Pthlh* mRNA respectively was confirmed in both WT and KO cultures before tamoxifen treatment (*Figure 1D*, *Figure 1—figure supplement 2A–B*). The p53-WT cells were largely unaffected by the shRNA's independent of tamoxifen treatment, except for an initial delayed proliferation in sh*Creb1* cultures (*Figure 1E–F*). The control (shLuc) infected KO osteoblasts hyperproliferated following tamoxifen treatment from day 15 onward (*Figure 1E–F*). In contrast, knockdown of either *Creb1* or *Pthlh* completely prevented the hyperproliferation of the KO +

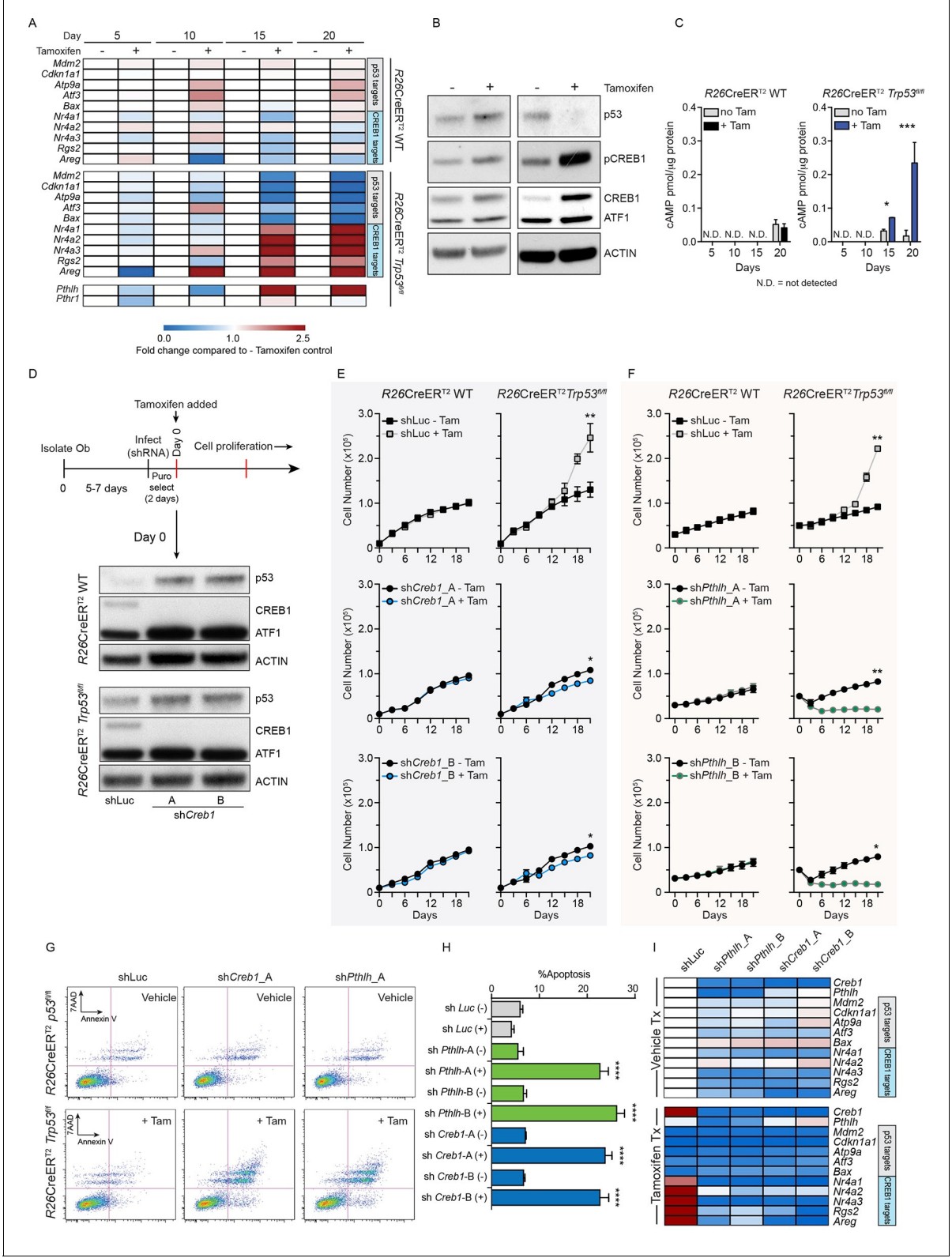

**Figure 1.** Intact PTHrP and CREB1 are necessary for hyperproliferation of p53-deficient primary osteoblasts. (**A**) Heat map of qPCR data. Expression of the PTHrP/CREB1 and p53 target genes between indicated cell types. Data from >3 independent cell lines for each, expressed as fold change relative to non-tamoxifen treated isogenic culture. (**B**) Western blot of p53, pCREB1 and CREB1, β-ACTIN used as a loading control. Data representative of 2–3 independent cell lines from each. (**C**) Quantification of cAMP levels (+IBMX) in the $R26$-CreER$^{T2}$$p53^{+/+}$ (vehicle and tamoxifen treated) and $R26$-

*Figure 1 continued on next page*

Figure 1 continued

CreER$^{T2}$p53$^{fl/fl}$ vehicle and tamoxifen treated (p53△/△) primary osteoblasts, and day 5, 10, 15 and 20 days post tamoxifen. Data from 2 *R26-*
*CreER*$^{T2}$p53$^{+/+}$ and 4 *R26*-CreER$^{T2}$p53$^{fl/fl}$ independent cultures; mean ± SEM. (D) Experimental outline for proliferation assay; Western blot of p53,
pCREB1 and CREB1 in indicated cell types, *β*-ACTIN used as a loading control at Day 0 of culture. Proliferation assays performed in the indicated
genotype post CREB1 (E) and PTHrP (F) knockdown with tamoxifen treatment commencing at day 0; shLuc = control shRNA; Data from 4 independent
*R26*-CreER$^{T2}$p53$^{fl/fl}$ and 2 *R26*-CreER$^{T2}$p53$^{+/+}$ cultures; mean ± SEM and statistics = area under the curve across the time course. (G) AnnexinV/7-AAD
profiles of *R26*-CreER$^{T2}$p53$^{fl/fl}$ +/- tamoxifen treatment infected with control (shLuc), sh*Creb1*_A or sh*Pthlh*_A. (H) Percent apoptotic cells in each culture
+/- tamoxifen. (I) Heat map of qPCR data. Expression of the p53 and PTHrP/CREB1 target genes between cell types; 3 independent cell cultures for
each condition. Data expressed as mean ± SEM (n=3). For all panels: *p<0.05, **p<0.01, ***p<0.001. See *Figure 1—figure supplement 1* and
*Figure 1—figure supplement 2*.

The following figure supplements are available for figure 1:

**Figure supplement 1.** Expression of p53 and PTHrP/CREB1 target genes in R26-CreER p53+/+ and R26-CreER p53fl/fl cultures +/- tamoxifen.

**Figure supplement 2.** Effects of shRNA against *Pthrp* and *Creb1* in R26-CreER p53+/+ and R26-CreER p53fl/fl cultures +/- tamoxifen.

tamoxifen cells (*Figure 1E–F*, *Figure 1—figure supplement 2C–F*). Therefore, intact PTHrP and
CREB1 signaling are required for the hyperproliferation of p53 deficient osteoblasts.

Finally, to assess the requirements for PTHrP and CREB1 in p53-deficient osteoblasts, we infected
cells with the respective shRNAs after they had been cultured for 21 days with tamoxifen, such that
the cells had already undergone the hyperproliferative transformation prior to knockdown. The p53-
KO osteoblasts underwent apoptosis within 48 hr of knockdown with either sh*Creb1* or sh*Pthlh*
whilst the isogenic control (-tam) cultures were minimally affected (*Figure 1G–H*). The knockdown of
PTHrP/CREB1 led to an expected downregulation of PTHrP-CREB1 targets (*Figure 1I*, *Figure 1—
figure supplement 2G–H*). The hyperproliferative effect of loss of *Trp53* in osteoblastic cells
required PTHrP and CREB1.

## Autocrine PTHrP is a primary stimulus of cAMP in OS

Having established the necessity of PTHrP and CREB1 for the hyperproliferation and survival of p53-
deficient osteoblasts, we sought to understand the contribution of this pathway in OS. We systemat-
ically profiled the contribution of PTHrP, cAMP and CREB1 in primary cell cultures derived from
murine OS models compared to primary osteoblasts. We made use of the *Sp7(Osx)*-Cre *Trp53*$^{fl/}$
$^{fl}$*Rb1*$^{fl/fl}$ model (Cre:lox deletion of *Trp53* and *Rb1*; referred to as fibroblastic OS) which yields a OS
characterised by predominant areas of fibroblastic or poorly differentiated/undifferentiated
(*Berman et al., 2008*) histology and a cell surface phenotype consistent with immature osteoblasts
(*Walkley et al., 2008*; *Mutsaers et al., 2013*). The second model was the *Sp7(Osx)*-Cre
TRE_shp53.1224*Rb1*$^{fl/fl}$ model (shRNA knockdown of *Trp53*; referred to as osteoblastic OS) which
histologically resemble osteoblastic OS with large mineralized areas, appreciated by von Kossa stain-
ing or microCT, and a cell surface phenotype of mature osteoblasts (*Mutsaers et al., 2013*). The
early passage cells from both models have comparable genetic and pharmacological sensitivities to
those of primary human patient derived OS cultures where tested (*Gupte et al., 2015*; *Baker et al.,
2015*). As a control population (referred to herein as "primary osteoblasts"), we isolated osteoblastic
cells from the collagenase digested long bones of wild-type C57BL/6 mice. These cells are negative
for haematopoietic markers (lineage markers, CD45, CD11b, F4/80), negative for the endothelial cell
surface marker CD31 and co-express CD51 and Sca-1. The majority of the cells have a cell surface
phenotype consistent with pre-osteoblasts (lin$^-$CD45$^-$CD31$^-$CD51$^+$Sca1$^+$) when the cultures are initi-
ated, and when induced to differentiate acquire a mature osteoblast/osteocyte gene expression
profile.

We first assessed PTHrP given that PTHrP stimulates cAMP generation following activation of
PTHR1, ultimately leading to CREB1 phosphorylation and transcriptional activation (*Figure 2A*).
Osteoblastic OS cells expressed high levels of *Pthlh* transcript (*Figure 2B*), consistent with our previ-
ous data identifying substantial levels of intracellular PTHrP in OS cells (*Ho et al., 2015*). As other
GPCRs are expressed on OS cells, such as β-adrenergic receptors (*Figure 2—figure supplement
1A*), we sought to determine if PTHrP was an OS autocrine ligand. Cells were treated with the

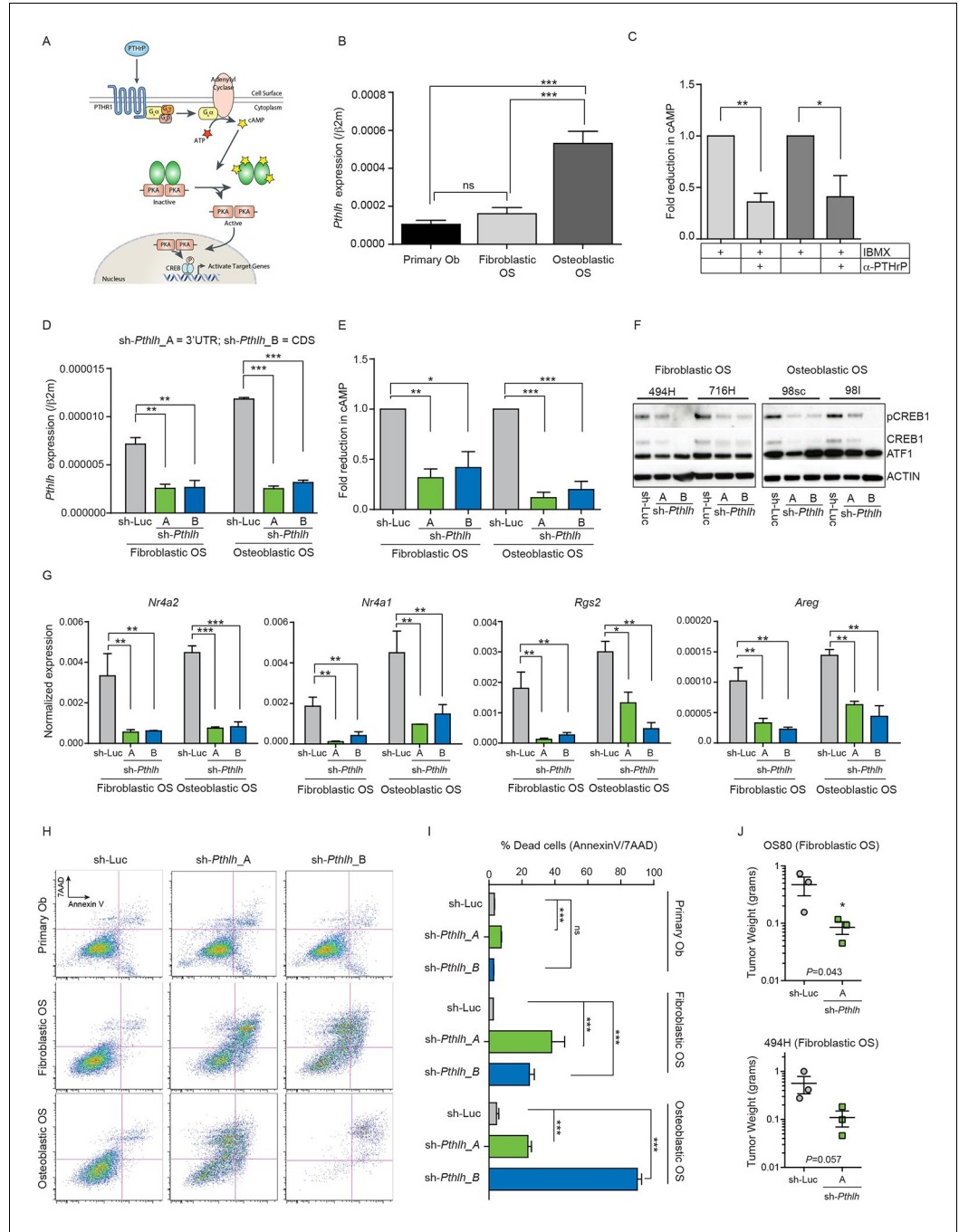

**Figure 2.** Cell autonomous stimulation of cAMP by PTHrP in OS. (**A**) Cartoon of PTHrP-PTHR1-cAMP-CREB1 axis. (**B**) qPCR expression of *Pthlh* normalized to *β2m*; mean ± SEM (n=3). (**C**) cAMP levels after anti-PTHrP antibody treatment for fibroblastic OS (light grey) and osteoblastic OS (dark grey). Expressed as normalized mean cAMP ± SEM ((n=3/subtype). (**D**) Knockdown of *Pthlh* transcript using 2 independent shRNA (**A** and **B**) in indicated OS subtypes. Data normalized to *β2m*, expressed as mean ± SEM (n=3/subtype). (**E**) Fold reduction of cAMP levels in sh-*Pthlh* infected OS subtype cells. IBMX in all treatments, data displayed as normalized mean cAMP ± SEM (n=3/subtype). The data is the mean of 3 independent cell cultures for each subtype. (**F**) pCREB1/CREB1 protein levels following knockdown of PTHrP. *Pan-ACTIN/*ATF-1 used as a loading control. Data are representative of 2 independent cell cultures from each OS subtype. (**G**) Expression of indicated CREB1 target gene transcripts following *Pthlh* knockdown. Means ± SEM (n=3/subtype). (**H**) AnnexinV/7-AAD staining of indicated cells following infection with two independent sh-*Pthlh* (A and B) or sh-Luc control. (**I**) Quantitation of dead cells in indicated cell type. The data represents 3 independent cell cultures for each type, mean ± SEM (n=3). *p<0.05, **p<0.001,

*Figure 2 continued on next page*

*Figure 2 continued*

***p<0.0001. (J) In vivo bilateral grafts of independent fibrobastic OS lines OS80 and 494H with control (sh-Luc) on one flank and sh-*Pthlh*_A on the other flank. Data expressed as mean weight ± SEM (n=3 tumours per shRNA per cell line; performed once); *P* value as indicated. See *Figure 2—figure supplement 1* and *Figure 2—figure supplement 2*.

The following figure supplements are available for figure 2:

**Figure supplement 1.** PTHrP, an endogenous ligand for cAMP signaling in OS.

**Figure supplement 2.** PTHrP overexpression alone does not initiate OS.

phosphodiesterase inhibitor, IBMX, without adding exogenous PTHrP, thus assaying the cAMP induced by autocrine activation of receptor-linked adenylyl cyclase by ligand(s) provided by the OS cells. Treatment with a neutralising anti-PTHrP antibody significantly and substantially reduced cAMP levels (*Onuma et al., 2004*) (*Figure 2C*, *Figure 2—figure supplement 1B*). Using the shRNAs against *Pthlh*, a >50% reduction in the cAMP accumulation was observed (*Figure 2D–E*, *Figure 2—figure supplement 1C*). The reduction of cAMP levels by *Pthlh* knockdown or by antibody mediated PTHrP neutralization are consistent with OS–derived PTHrP as an endogenous ligand promoting cAMP accumulation.

Knockdown of *Pthlh* (*Figure 2—figure supplement 1D*) reduced the proliferation (*Figure 2—figure supplement 1E*), transcription of known target genes (*Figure 2—figure supplement 1F–G*) and the levels of pCREB1, and surprisingly, total CREB1 in OS cells at early time points post infection (*Figure 2F*). Correspondingly there was a significant reduction in the basal expression of CREB1 target genes (*Figure 2G*). Next, cell survival 48–72 hr after shRNA infection was assessed. Primary osteoblasts were largely unaffected by *Pthrp* knockdown. In contrast there was a rapid induction of apoptosis following *Pthlh* knockdown in OS cells (*Figure 2H–I*). To determine the contribution of elevated PTHrP expression on OS initiation, we retrovirally overexpressed PTHrP in wild-type primary osteoblasts. Surprisingly, the cells overexpressing high levels of PTHrP failed to thrive and a significant proportion underwent cell death indicating that PTHrP overexpression alone is not sufficient to support OS initiation (*Figure 2—figure supplement 2A–C*). Two different fibroblastic OS lines infected with control (sh-Luc) or sh-*Pthlh*_A were grafted subcutaneously in vivo and both had significantly reduced proliferation as measured by tumor weight (*Figure 2J*), comparable to the effects of sh*Pthr1* knockdown in the same OS lines (*Ho et al., 2015*). These results demonstrate that PTHrP is a critical, OS cell-derived stimulus of the elevated cAMP in OS.

## Elevated cAMP levels in OS lead to sustained CREB1 activation

We next assessed basal cAMP levels in normal osteoblasts and the two OS subtypes in unstimulated proliferating cultures (with and without phosphodiesterase inhibition but no exogenous ligand treatment). OS cells produced significantly greater amounts of intracellular cAMP compared to primary murine osteoblasts (*Figure 3A*, *Figure 3—figure supplement 1A*). Furthermore, treatment with the direct cAMP agonist forskolin in the presence of IBMX resulted in an increased and sustained accumulation of cAMP in OS compared to primary osteoblasts (*Figure 3B*). Without IBMX the relative responses to forskolin remained the same, albeit with lower cAMP levels (*Figure 3—figure supplement 1B*).

Based on the elevated cAMP in OS, we tested the dynamics of CREB1 phosphorylation in serum starved cells to acute elevation of cAMP induced by forskolin. Induction of pCREB1 was rapid in normal osteoblasts, peaking at 30 min, then reducing throughout the 120 min time course as expected (*Figure 3C–D*). In contrast, OS cells displayed continuous and persistent activation of pCREB1, consistent with the cAMP levels (*Figure 3C–D*). The OS-specific altered dynamics of cAMP and pCREB1 resulted in aberrantly extended transcriptional activation of known CREB1 target genes based on both transcript expression and chromatin occupancy of CREB1/pCREB1 (*Figure 3E–F*, *Figure 3—figure supplement 1C–D*). The requirement for CREB1 in the transcription of these targets in fibroblastic OS was confirmed using siRNA (*Figure 3—figure supplement 1E*). Importantly, there was no

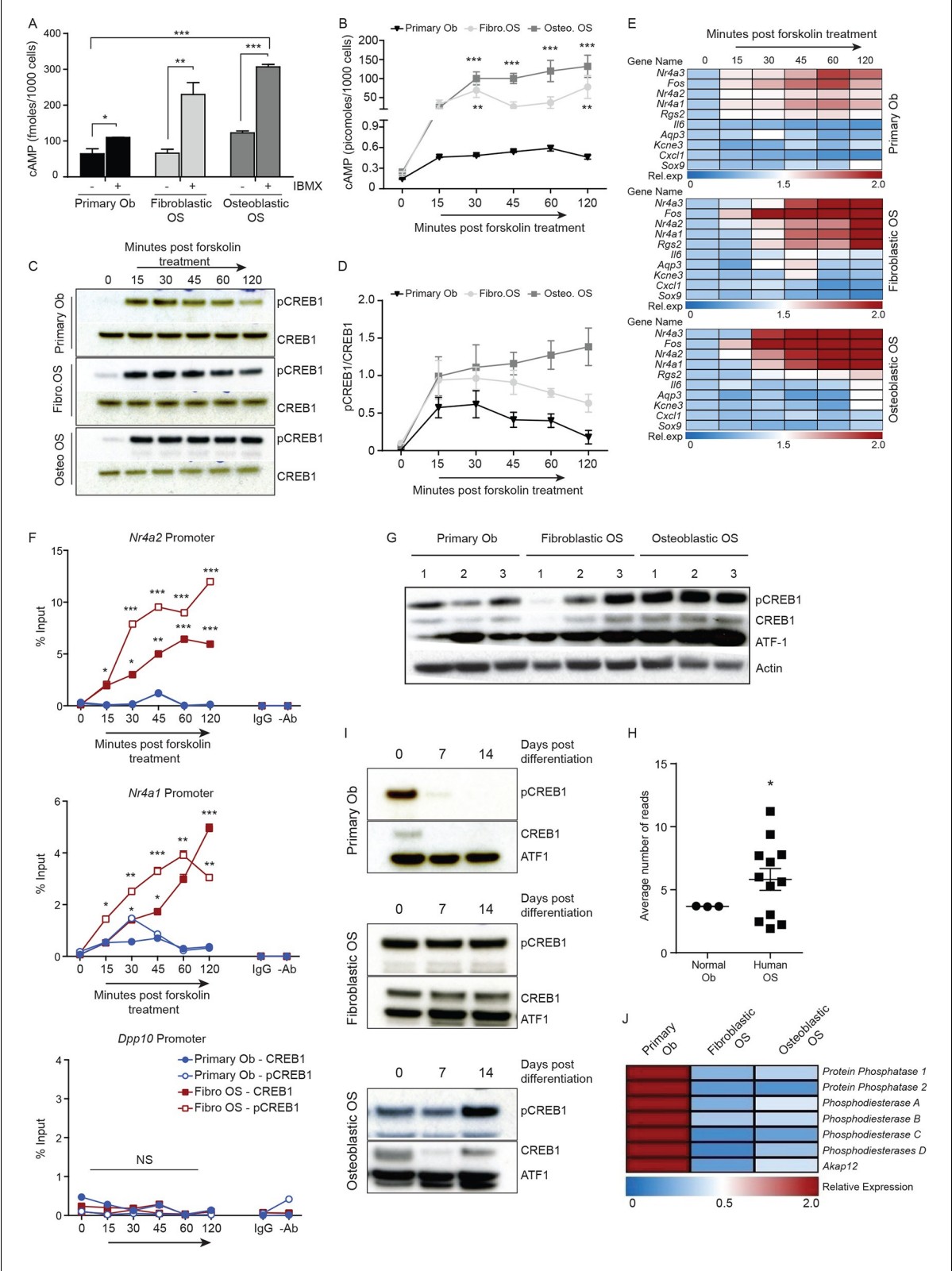

**Figure 3.** Persistent, elevated cAMP production in OS compared to primary osteoblasts. (**A**) cAMP levels in indicated cells (1000 cells per well) with and without IBMX treatment. Data from 3 independent cultures per type, mean ± SEM. (**B**) Intracellular cAMP levels in indicated cell type following treatment with forskolin; mean ± SEM (n=3 per cell type; 1000 cells per well); statistical significance for OS vs normal Ob; all points of fibroblastic vs osteoblastic OS not significantly different. (**C**) Western blot and (**D**) quantification of CREB1/pCREB1 during a time course of cAMP activation by

*Figure 3 continued on next page*

*Figure 3 continued*

forskolin. Data representative of 2 independent cultures each. (**E**) Heat map of qPCR data. CREB1 target gene expression in indicated cells; data expressed as relative expression. (**F**) ChIP analysis of CREB1/pCREB1 on the promoters of indicated genes over a 2 hr time course following stimulation with forskolin. Data is represented as percentage of input. The data from 2 independent cell lines for each subtypes mean occupancy ± SEM (n=2–3 assays per line). (**G**) Western blot of CREB1/pCREB1 expression in proliferating non stimulated cultures, *β*-ACTIN used as a loading control. Data representative of 3–4 independent cell lines from each type. (**H**) *CREB1* transcript expression in human osteoblasts and osteosarcoma (data taken from PMID: 25961939). (**I**) Western CREB1/pCREB1 expression in indicated cell types under differentiative conditions, ATF-1 used as a loading control. (**J**) Relative expression of negative regulators of cAMP in OS subtypes compared to primary osteoblasts by qPCR and normalized to *β2m* represented as a heat map (n=3/cell type). *p<0.05, **p<0.001, ***p<0.0001. See *Figure 3—figure supplement 1* and *Figure 3—figure supplement 2*.

The following figure supplements are available for figure 3:

**Figure supplement 1.** cAMP is constitutive in mouse OS leading to continuous phosphorylation of CREB1.

**Figure supplement 2.** Altered *Creb1* dynamics in osteoblasts and OS.

evidence of compensation for loss of *Creb1* by the related *Crem1* in either OS subtype (*Figure 3—figure supplement 1F–G*) (*Mantamadiotis et al., 2002*).

## OS cells fail to reduce CREB1 activity during maturation

We assessed the levels of pCREB1, the downstream transcriptional effector of cAMP signaling, in proliferating OS cells compared to primary osteoblasts. CREB1 was more prominently phosphorylated in the osteoblastic OS than in either the fibroblastic OS or primary osteoblasts (*Figure 3G*). qRT-PCR using independent OS cultures and primary osteoblasts demonstrated that the mean *Creb1* expression was 2–3 fold higher in osteoblastic OS compared to fibroblastic OS and primary osteoblasts (*Figure 3—figure supplement 1H*). Analysis of RNA-seq from human OS revealed a significant increased in the expression level of *Creb1* in OS compared to normal osteoblasts (*Figure 3H*) (*Moriarity et al., 2015*). During culture in differentiation inductive conditions, CREB1 levels reduced over the first 7 days in osteoblasts and stayed low for the remainder of the culture (*Figure 3I*, *Figure 3—figure supplement 2A*). In contrast, OS cells maintained CREB1 expression under the same conditions (*Figure 3I*, *Figure 3—figure supplement 2B*). The decrease in CREB1 expression (both transcript and protein levels) upon differentiation was confirmed in primary human osteoblasts (*Figure 3—figure supplement 2C–D*).

In human OS, somatic SNV mutations in negative regulators of cAMP levels were described, including members of the phosphodiesterases (PDE), A kinase anchoring proteins (AKAP) and protein phosphatases (PP) (*Chen et al., 2014*). There was a 2–3 fold decreased expression of several members of these gene families in the murine OS cells compared to primary osteoblasts (*Figure 3J*, *Figure 3—figure supplement 2E*). The reduced expression of PDE, AKAPs and PPs would be expected to favour the accumulation and action of cAMP following GPCR activation.

## Constitutively active cAMP differentially impacts primary osteoblasts and p53-deficient OS

As intracellular cAMP increased following p53 deletion in primary osteoblasts, we sought to determine the effect of elevated cAMP levels on normal osteoblast differentiation. Primary osteoblasts were treated with the forskolin and their response compared to that of OS cells (*Walkley et al., 2008*; *Mutsaers et al., 2013*). Forskolin stimulates cAMP independently from cell surface GPCRs so was used instead of PTHrP, allowing a meaningful comparison of the isolated consequences of elevated cAMP as undifferentiated primary osteoblasts express less PTHR1 compared to the OS cells (*Mutsaers et al., 2013*).

After 72 hr treatment, primary osteoblasts had altered cell surface phenotypes and reduced expression of *Runx2* and *Osx* (*Figure 4A–C*) (*Mutsaers et al., 2013*). Brief exposure to forskolin (24 hr) yielded the same result (*Figure 4—figure supplement 1A–B*). During differentiation, cAMP activation led to decreased expression of differentiation markers and failure to normally mineralise (*Figure 4D–E*, *Figure 4—figure supplement 1C–D*). Therefore continuously elevated cAMP increased features associated with immature osteoblasts. In OS cells, the cell surface phenotypes

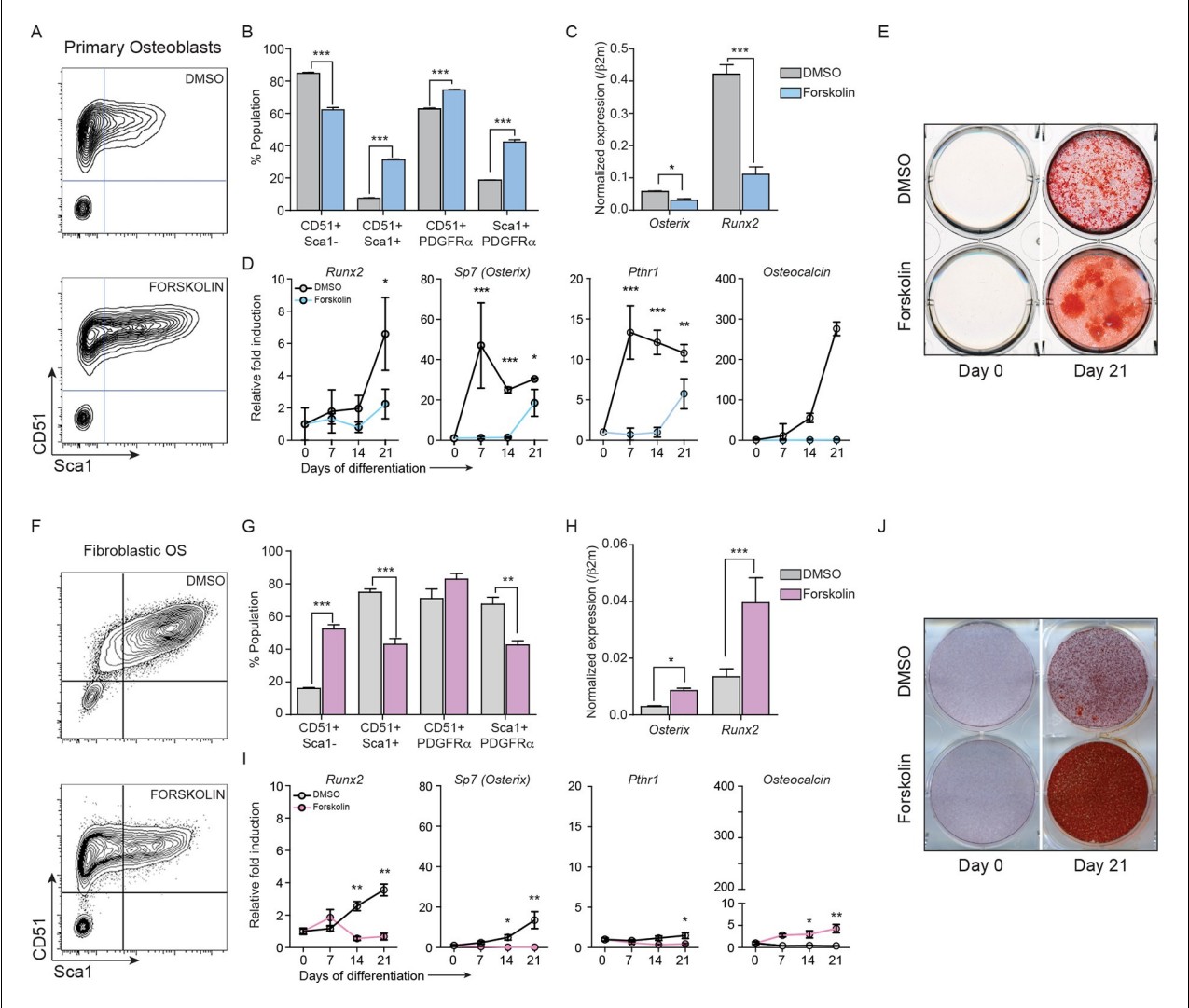

**Figure 4.** Constitutively elevated cAMP differentially affects primary osteoblasts and osteosarcoma cells. (**A**) Primary osteoblasts treated with DMSO or forskolin for 72 hr and assessed for expression of Sca-1, CD51, PDGFRα , representative results shown, n=3 independent experiments. (**B**) Quantitation of cell surface markers from each treatment (n=3 independent cultures) (**C**) Expression of *Sp7 (Osterix) and Runx2* by qPCR after 72 hr of forskolin treatment. Expression levels normalized to *β2m*; mean ± SEM (n=3). (**D**) Expression level of indicated genes over 21 days of treatment with DMSO or forskolin. Expression normalized to *β2m;* mean ± SEM (n=3). (**E**) Mineralisation analysis of primary osteoblasts at day 21 after treatment. Images are representative of 3 independent experiments. (**F**) Fibroblastic OS cells were treated with DMSO or forskolin for 72 hr and assessed for expression of Sca-1, CD51, PDGFRα, representative results shown. (**G**) Quantitation of cell surface markers from (n=3 independent cultures of fibroblastic OS) from each treatment. (**H**) Expression of *Sp7 (Osterix) and Runx2* in fibroblastic OS by qPCR following 72 hr treatment. Expression levels normalized to *β2m*; mean ± SEM (n=3). (**I**) Expression of indicated genes in fibroblastic OS over 21 days from each treatment. Expression normalized to *β2m;* mean ± SEM (n=3). (**J**) Representative images of alizarin red stained fibroblastic OS cells treated with DMSO or forskolin for 21 day; n=3 independent OS cultures; *p<0.05, **p<0.001, ***p<0.0001. See *Figure 4—figure supplement 1* and *Figure 4—figure supplement 2*.

The following figure supplements are available for figure 4:

**Figure supplement 1.** cAMP has different effects in OS and primary osteoblasts.

**Figure supplement 2.** Level of cAMP in primary osteoblasts and osteosarcoma cells +/- forskolin.

and expression of *Runx2* and *Sp7* were the inverse of primary osteoblasts after 72 hr forskolin treatment (*Figure 4F–H*, *Figure 4—figure supplement 1E*). Under differentiation conditions, forskolin induced less profound changes in the expression of markers of osteoblast maturation, with the

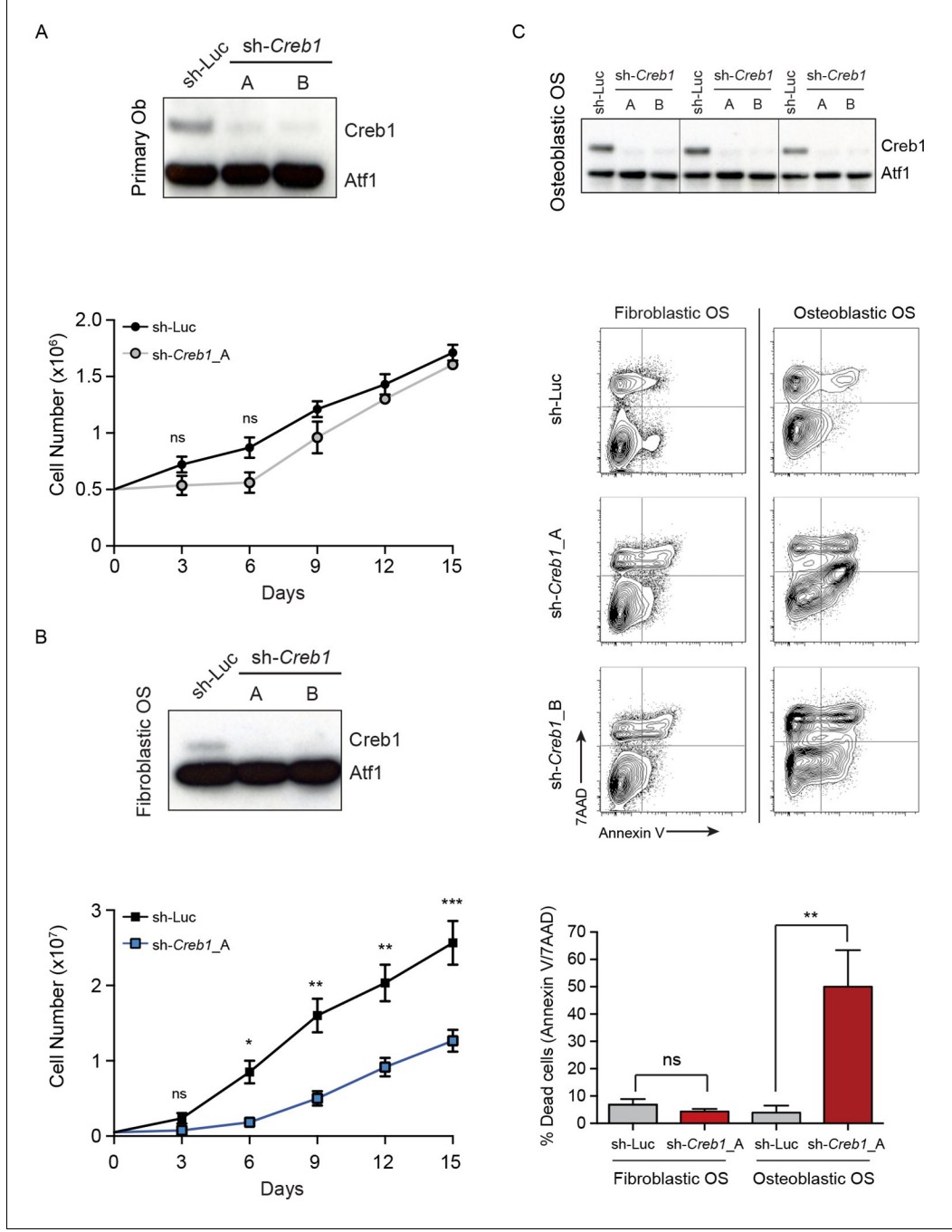

**Figure 5.** CREB1 is differentially required for proliferation and survival by OS subtypes. (**A**) CREB1 in primary osteoblasts 72 hr after infection with indicated shRNA construct. ATF1 was used as a loading control; representative blot from 3 independent cultures; proliferation plotted as mean ± SEM (n=3). (**B**) Western blot of CREB1 and proliferation kinetics of sh*Creb1* knockdown and sh-Luc fibroblastic OS; representative blot from 3 independent OS lines; proliferation as mean ± SEM (n=3). (**C**) Western of CREB1 in osteoblastic OS cells 72 hr after infection; ATF1 = loading control. Viability (annexinV/7AAD) of indicated OS subtype following infection with each shRNA. Data are representative of 3 independent cell lines/type; quantitation of dead cells. Data from 3 independent cell lines/subtypes; mean ± SEM. *p<0.05, **p<0.001, ***p<0.0001

exception of *Osteocalcin* (*Figure 4I*), and resulted in increased mineralization (*Figure 4J*, *Figure 4—figure supplement 1D–G*). To determine the consequences of CREB1 retention in OS cells, CREB1 was knocked down using both shRNAs (3'UTR, CDS) in fibroblastic OS cells and differentiation evaluated (*Figure 4—figure supplement 1J–L*). Both early and late markers of maturation were reduced

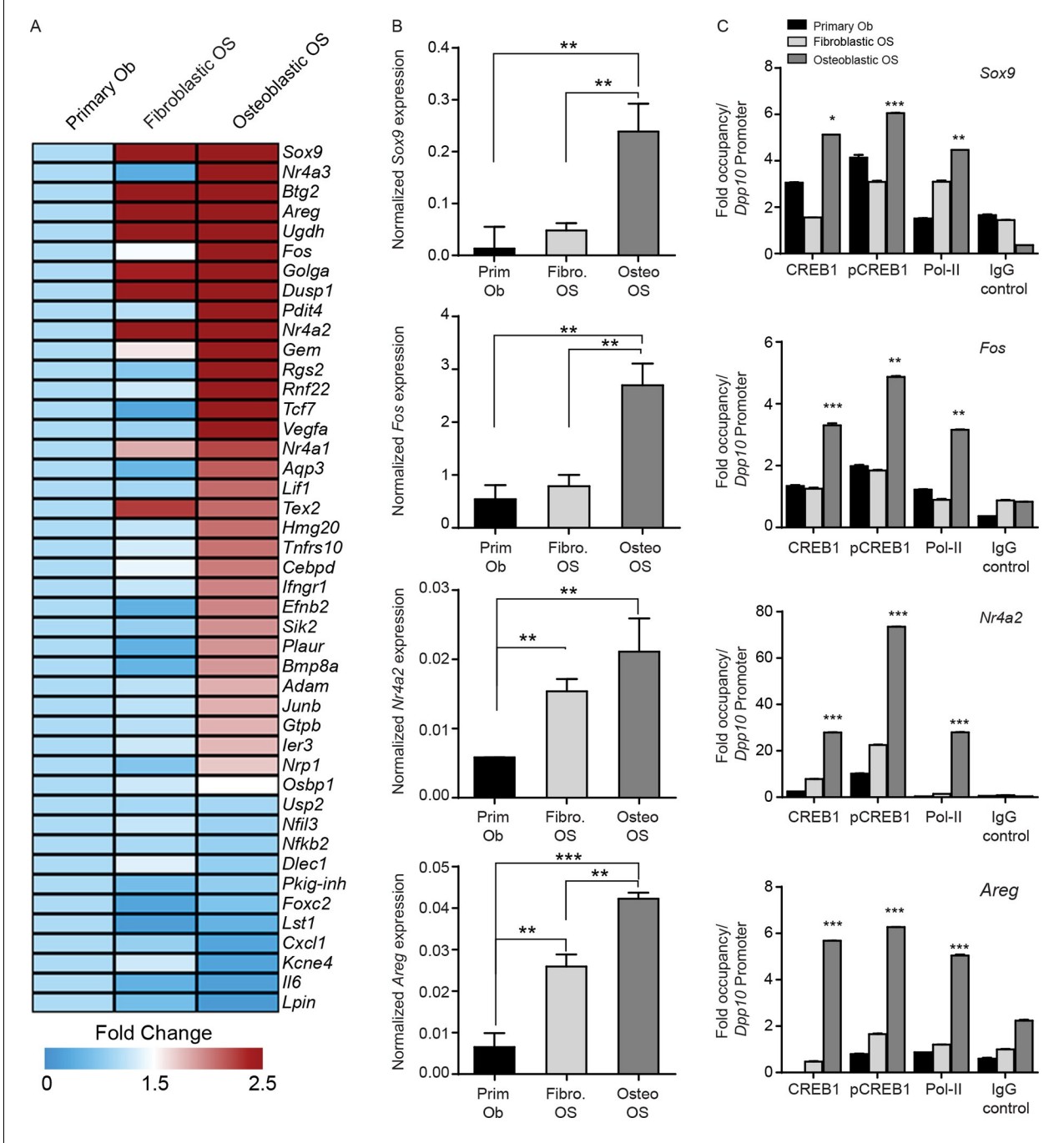

**Figure 6.** CREB1 signatures discriminate OS subtypes. (**A**) Heat map of qPCR data. Expression of the PTHrP/CREB1 gene set between indicated cell types. Data from 3 independent cultures for each, expressed as fold change relative to primary osteoblasts. (**B**) Examples of CREB1 target gene expression between the indicated cell types. Expression levels normalized to *β2m* and depicted as relative expression ± SEM (n=3). (**C**) ChIP-qPCR for the indicated target genes from proliferating cells (no exogenous ligand/stimulus of cAMP applied) with CREB1, pCREB1 and pPolII. Data represented as fold occupancy relative to *Dpp10* promoter, expressed as mean ± SEM. *p<0.05, **p<0.001, ***p<0.0001. See also *Figure 6—figure supplement 1*.

The following figure supplement is available for figure 6:

**Figure supplement 1.** CREB1 defines OS subtypes by driving specific gene signatures.

with sh-*Creb1* (*Figure 4—figure supplement 1J*). Mineralization was significantly reduced in sh-*Creb1* expressing cells compared to controls (*Figure 4—figure supplement 1K–L*). The level of intracellular cAMP achieved with forskolin treatment is significantly higher that that achieved by cell derived autocrine/paracrine stimuli, such as secreted PTHrP (*Figure 4—figure supplement 2A–C*). These levels likely reflect maximal stimulation through the cAMP pathway in these cells which yields a distinct biological effect on primary osteoblastic cells compared to OS derived primary cultures. Therefore continuously elevated cAMP has distinct effects on the behaviour of normal osteoblasts and OS.

## OS subtypes have a differential dependence upon CREB1

Reducing PTHrP levels caused apoptosis of OS cells and also reduced levels of CREB1/pCREB1 (*Figure 2F*, *Figure 2H–J*). To determine if loss of CREB1 impacted OS cell survival similarly we assessed the effects of *Creb1* knockdown. In all cohorts there was loss of CREB1 protein (*Figure 5A–C*). CREB1 knockdown in primary osteoblasts caused reduced proliferation in the first week then the cells recovered and proliferated similarly to control infected cells thereafter, with no apparent effect on survival (*Figure 5A*). Loss of CREB1 in fibroblastic OS cells resulted in sustained proliferation impairment, yet cell survival was not appreciably impacted (*Figure 5B*). Knockdown of CREB1 in osteoblastic OS, the most common clinical subtype, caused profound proliferation arrest and apoptosis (*Figure 5C*). The effect was so complete that we have not been able to establish stable sh-*Creb1* expressing cultures from the osteoblastic OS. The phenotype was observed with both sh-*Creb1* constructs and in ≥3 independent cultures. Therefore, CREB1 is dispensable for normal osteoblast function yet required for proliferation of fibroblastic OS and survival of osteoblastic OS.

## OS subtypes can be discriminated by CREB1 target gene signatures

Given the subtype specific effects of CREB1 knockdown, we sought to determine if CREB1 gene signatures could be used to appreciate differences between the subtypes. We modelled our analysis on the evidence that PTHrP was the endogenous ligand leading to cAMP accumulation and CREB1 activation. We defined a PTHrP-specific gene signature bioinformatically from previous microarrays comparing PTHrP(1–141) to PTH(1–34) in differentiating osteoblasts (*Allan et al., 2008*). The top 45 candidates from the signature were validated. Using proliferating primary osteoblasts, fibroblastic OS and osteoblastic OS cells, 32 of the 45 genes were most highly expressed in the osteoblastic OS (*Figure 6A*). Archetypal CREB1 target genes were all more highly expressed in osteoblastic OS (*Figure 6B*, *Figure 6—figure supplement 1A–C*). Chromatin immunoprecipitation-PCR (ChIP-qPCR) demonstrated enrichment of active pCREB1 and serine 2 phosphorylated RNA polymerase II (pPolII), a mark of active transcription, on CREB1 target genes in osteoblastic OS (*Figure 6C*, *Figure 6—figure supplement 1D*) (*Ho and Shuman, 1999*). Promoter binding was generally lower in the fibroblastic OS, consistent with the expression patterns of the target genes. The enhanced binding of CREB1 in osteoblastic OS corresponded to the elevated levels of cAMP and osteoblastic OS is characterised by increased CREB1 activity.

## Somatic SNV mutations in human OS overlap with the cAMP-CREB1 interactome

Mutations and oncogenic effects of the cAMP pathway have been described in the context of other tumor types, including breast (*Kok et al., 2011*; *Miller, 2002*; *Beristain et al., 2015*; *Pattabiraman et al., 2016*) and haematological malignancies (*Pigazzi et al., 2013*; *Sandoval et al., 2012*; *Shankar et al., 2005*; *Smith et al., 2005*; *Mulligan et al., 2011*) amongst other tumors. The mutational landscape of OS has been recently defined, identifying 1704 somatic single nucleotide variations (SNV mutations) across 20 cases of sporadic conventional OS (*Chen et al., 2014*). To determine if these somatic SNV mutations were functionally related, we assessed pathways enriched within the somatic SNV mutations (*Figure 7A*). In the top 20 pathways were signatures associated with ion channel complexes, transmembrane transporter complexes, PI3K signaling, calcium channel signaling, and protein kinase A (PKA) activity, all of which are related to cAMP (*Figure 7A*). Based on this result, we compared the somatic SNV mutations of human OS to the 169 genes comprising the KEGG cAMP interactome. To determine if this was OS specific or a more generalised feature of tumors, we further compared the cAMP interactome with whole genome sequencing that identified

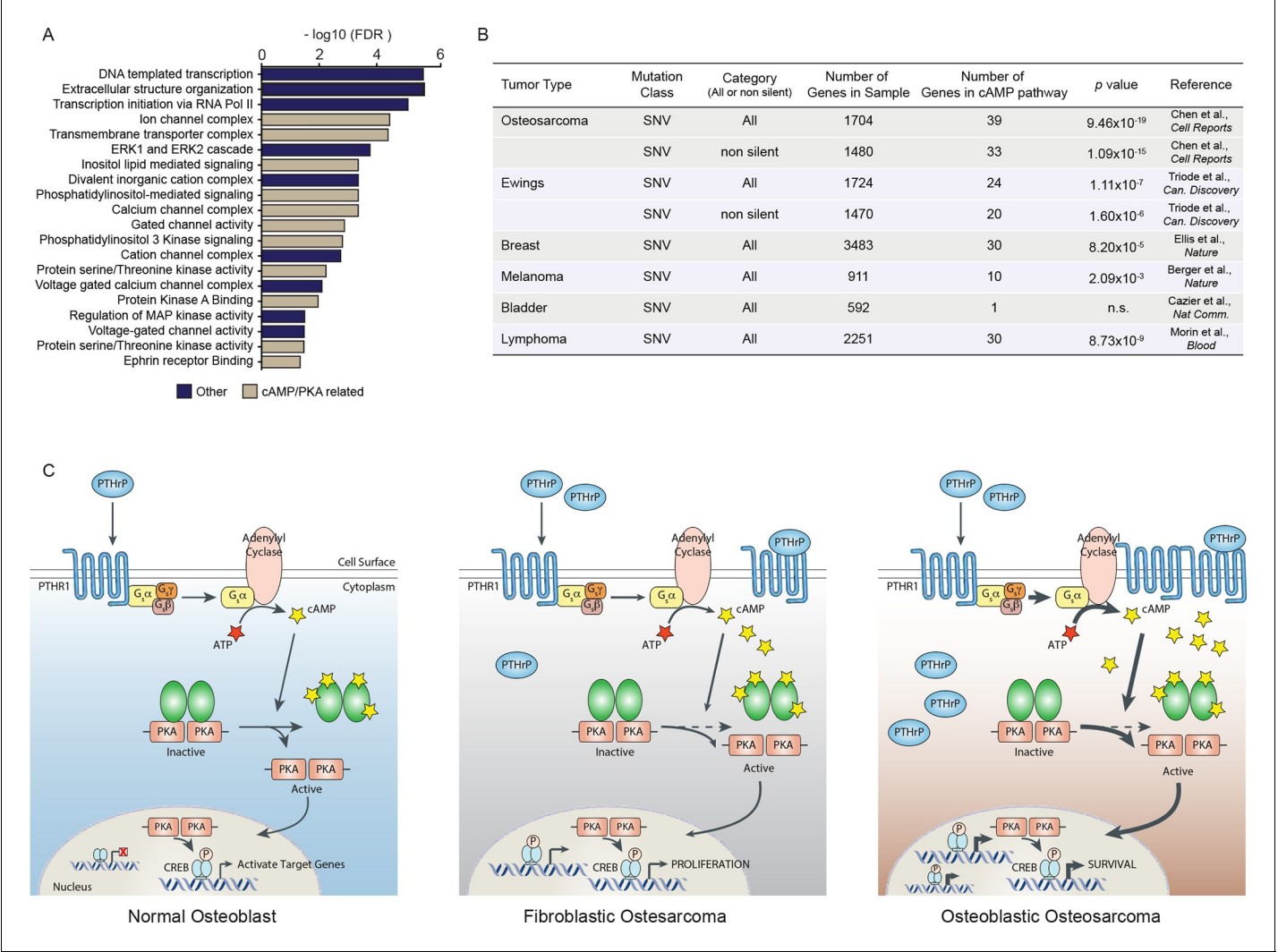

**Figure 7.** A high proportion of the cAMP interactome are somatic SNV mutations in human osteosarcoma. (**A**) Analysis of functional pathways within the somatic SNV mutations of human OS using Cytoscape. Brown color indicates a cAMP related pathway, blue color indicates non cAMP related pathways. (**B**) Analysis of the enrichment for somatic SNV mutations within the cAMP interactome in each of the indicated tumor types. Based on somatic SNV mutations identified by whole genome sequencing. *P* value defined using hypergeometric distribution test. (**C**) Graphical summary of the differences between primary osteoblasts and OS subtypes regarding cAMP and CREB1 function. See also *Figure 7—figure supplement 1*.

The following figure supplement is available for figure 7:

**Figure supplement 1.** Analysis of cAMP and cGMP pathway enrichment of somatic SNV tumor mutations.

SNV mutations of other human cancers (*Ellis et al., 2012*; *Morin et al., 2013*; *Berger et al., 2012*; *Cazier et al., 2014*; *Tirode et al., 2014*). Genes within the cAMP interactome were most highly enriched within the somatic SNV mutations of human OS (*Figure 7B*). These data suggest that, despite the diverse mutational pattern of somatic SNV mutations in human OS, recurrent and enriched changes in the cAMP and CREB1 pathways occur in OS.

## Discussion

There remains an incomplete understanding of the cellular rewiring that accompanies the loss or null mutation of *TRP53*. Although the p53 pathway can be targeted, most interventions aim to activate or restore function of the mutant P53 protein, an approach not feasible in null settings (*Khoo et al., 2014*), the most common case in OS (*Chen et al., 2014*). The identification of critical cellular

pathways that are activated/altered in response to *TRP53* deficiency may yield novel avenues to test therapeutically.

Using the fact that loss/mutation of *TRP53* is essentially universal in OS, we modelled an initiating lesion in primary osteoblastic cells and used this to understand the consequences in these cells. Several lines of evidence have implicated the cAMP pathway in OS, yet the functional requirement for this pathway has not been evaluated. We recently demonstrated a role for PTHR1 in OS proliferation and maintenance of the undifferentiated state (*Ho et al., 2015*). We therefore sought to determine if these pathways intersected in the p53 deletion dependent initiation and maintenance of OS. There was a co-ordinated increase in *Pthlh*, cAMP per cell and CREB1 levels and transcriptional activity when osteoblasts became functionally p53-deficient. This was unexpected, as it suggested that a very early event in the initiation of OS following the loss of p53 is the activation of the PTHrP→cAMP→CREB1 signaling axis. Whilst the detailed processes through which p53 loss activates this pathway is to be resolved, these results demonstrate that this is an essential pathway in the manifestations of the p53-deficient phenotype in osteoblastic cells. Furthermore, we demonstrated that this pathway was also required for the survival of osteoblasts rendered p53-deficient prior to loss of PTHrP or CREB1. Therefore, activation of the PTHrP→cAMP→CREB1 signaling axis appears to be a core component of the rewiring of osteoblastic lineage cells in response to loss of *Trp53* (*Figure 7C*).

Our results, together with prior studies implicating cAMP pathways in OS, indicate that elevated cAMP signaling could be considered oncogenic in OS. Forcibly increasing intracellular cAMP in normal osteoblasts retained them in an immature state, consistent with the recent report identifying forskolin as an inducer of pluripotency (*Hou et al., 2013*). Analysis of human OS identified somatic SNV mutations in a number of negative regulators of cAMP levels (*Chen et al., 2014*). Inactivating mutations in these would be predicted to elevate PKA and CREB1 activity. Khokha and colleagues identified mutations in *Prkar1a* using a murine model of OS, with a corresponding *PRKAR1A* low human OS subset defined (*Molyneux et al., 2010*). In the same study, mutually exclusive amplification of the α-subunit of PKA (*Prkarca*) was also reported. Our data reconcile these observations and indicate that the increased PKA-CREB1 activity is ultimately important for this tumor, in that it has evolved mechanisms ensuring elevated and persistent cAMP levels mediated primarily by autocrine production of PTHrP and modulated by reduced expression of negative regulators of cAMP activity. In normal physiology, PTHrP acts in a paracrine manner whilst in OS, as reported here, there is capacity for PTHrP to act in an autocrine and paracrine manner, as well as intracrine activities. Reducing PTHrP levels was tolerated by normal osteoblasts but not OS cultures.

While mutations promoting accumulation of cAMP are important, the stimulus of cAMP had not been defined. We demonstrate that PTHrP is a key OS cell-intrinsic inducer of cAMP. PTHrP is required for normal bone homeostasis via its actions upon osteoblastic cells (*Miao, 2005*). The functions of PTHrP in malignant osteoblast biology, however, have not been resolved. Reducing PTHR1 expression on OS cells enhanced differentiation and mineralization in vivo (*Ho et al., 2015*). There was a trend to greater levels of PTHrP, notably intracellular, in osteoblastic OS compared to fibroblastic OS (*Ho et al., 2015*). These data are consistent with the present measurements of pCREB1 levels in the subtypes and the CREB1 dependence of the osteoblastic OS. We did not previously impact OS in vivo using a neutralizing anti-PTHrP antibody (*Ho et al., 2015*). We reconcile the failure to achieve a therapeutic dose of antibody in vivo with high concentrations of PTHrP likely within the immediate cell environment in OS (*Ho et al., 2015*). The reduction in CREB1 levels when PTHrP was reduced was unexpected, raising the possibility to be explored, that PTHrP may contribute to maintenance of CREB1 levels in an analogous manner to the role of JAK2 in preventing proteasomal degradation of CREB1 (*Lefrancois-Martinez et al., 2011*). Coupled with the present data, targeting PTHrP is a candidate therapeutic strategy in OS, although any intracrine contribution of PTHrP needs to be evaluated. The therapeutic targeting of components downstream of PTHrP/PTHR1 signalling may also be feasible, with several recent reports of inhibitors of CREB signalling activity (*Mitton et al., 2016*; *Xie et al., 2015*), although these are yet to progress to preclinical evaluation.

The management of OS has not substantively changed for the last three decades. The identification of the pathways that are utilised during the evolution from osteoblastic lineage cells to OS cells may reveal new means to target this tumour. Recent sequencing and modeling has revealed the genetic complexity and diverse mutational patterns of OS, yet underlying these data is the recurrent and universal inactivation of the P53 pathway (*Chen et al., 2014*; *Moriarity et al., 2015*). The Notch

pathway has been implicated in OS and is potentially druggable, however the evidence from human OS is equivocal as mutations in this pathway are not common in sporadic OS (*Tao et al., 2014*). Recent work from two groups using whole genome screening identified the PI3K/mTOR pathway as a conserved therapeutic vulnerability in OS, demonstrating the power of understanding the biological networks underpinning OS (*Perry et al., 2014*; *Gupte et al., 2015*). The identification of pathways synthetically lethal with p53-deficiency, or that are required for the maintenance of p53 deficient phenotypes, will yield new means to target these cells. Using this approach we have defined a requirement within the osteoblast lineage for continuous PTHrP and CREB1 activity for the initiation and maintenance of OS, yet normal osteoblasts could tolerate the depletion of both of these factors. A striking feature of the requirement for CREB1 was the differences between OS subtypes. The OS subtypes could be discriminated from normal osteoblasts and each other based on the progressive enrichment of CREB1 gene signatures that reflected their dependence on this pathway for proliferation and survival. These observations raise the largely unexplored possibility that OS subtypes are at some level genetically distinct and have different biological dependences. Collectively, the constitutive activity of the PTHrP-cAMP-CREB1 axis, tightly coupled to loss of p53, represents an essential node in OS that is amenable to therapeutic inhibition at multiple levels.

## Materials and methods

### Study approval

All experiments involving animals were approved by the Animal Ethics Committee of St. Vincent's Hospital, Melbourne. Primary human osteoblasts were isolated from bone marrow aspirates from the posterior iliac crest of de-identified healthy human adult donors with informed consent and consent to publish (IMVS/SA Pathology normal bone marrow donor program RAH#940911a, Adelaide, South Australia).

### Animals

Balb/c nu/nu mice (ARC, Perth, WA) were used as recipients for transplant of OS cell lines. For in vivo tumor growth 25,000 (OS80) or 75,000 (OS494H) cells were implanted subcutaneously on the back flank of Balb/c nu/nu recipients (*Ho et al., 2015*). Cells were resuspended in extracellular matrix (Cultrex PathClear BME Reduced Growth Factor Basement Membrane Extract). All animals received sh-Luc infected cells on one flank and sh-*Pthlh* infected cells on the alternate flank.

### OS cell cultures and normal osteoblastic cells

Primary mouse OS cell cultures were derived from primary tumors from each of the two mouse models of OS and were maintained and studied for less than 15 passages (*Walkley et al., 2008*; *Mutsaers et al., 2013*) (all cells used were obtained directly from primary and metastatic tumor material isolated directly from genetically engineered mouse models of OS, no further authentication performed by the authors, mycoplasma negative as tested by PCR based assay by the Victorian Infectious Diseases Reference Laboratory). Normal mouse osteoblastic cells were derived from crushed and collagenase digested long bones of 8-wk old C57BL/6 mice. Osteoblastic cell populations were purified either by FACS (FACSAria, BD Biosciences, San Jose, CA) as previously described (*Semerad et al., 2005*; *Singbrant et al., 2011*), or directly cultured following collagenase digestion (all primary osteoblasts were isolated directly from mouse long bone, no further authentication performed by the authors, not tested for mycoplasma). All cell cultures were maintained in αMEM (Lonza, Basel, Switzerland) medium supplemented with 10% fetal bovine serum (Sigma, St. Louis, MO, USA; non heat inactivated), 2 mM Glutamax (Life Technologies, Carlsbad, CA, USA), and except in siRNA transfection experiments, 100 U/ml penicillin and streptomycin (Life Technologies) antibiotics. Cells were cultured in a humidified 5% $CO_2$ atmosphere at 37° Celsius.

Normal human osteoblasts were derived from bone marrow aspirates from the posterior iliac crest of healthy human adult donors (17–35 years of age), with informed consent (IMVS/SA Pathology normal bone marrow donor program RAH#940911a). The cells were outgrown from the bony spicules that were collected following filtration of the BM through a 70 μm filter (*Atkins et al., 2002*). The human osteoblasts were obtained directly from primary human bone material, no further authentication performed by the authors, not tested for mycoplasma.

## In vitro differentiation of OS cell lines, and primary osteoblasts

All cells were seeded at 3000 cells/cm$^2$ on 6-well plates in αMEM with 10% FCS three days prior to differentiation induction. When cells had reached 100% confluence (Day 0), control cells were harvested, and all other cells were replenished 3 times per week with osteoblastic differentiation media: αMEM (Lonza), 10% (v/v) FBS, 25mM HEPES, 1% (v/v) (Gibco), Penicillin-Streptomycin (Gibco), 2 mM GlutaMAX (Gibco), 50 µg/ml ascorbate (Sigma), 0.01 M β-glycerophosphate (Sigma).

## cAMP response assays

Three independent cell lines from each group were used. 500 and 1000 cells from each group were seeded in triplicates in a 384 well plate. Cells were treated for 1 hr +/-100 µM Isobutylmethylxanthine (IBMX) before measuring intracellular cAMP. Intracellular cAMP was measured using the LANCE ultra cAMP kit (Perkin Elmer, AD0262) as directed by the manufacturer. Kinetics of cAMP was demonstrated using 10 µM forskolin over a time course of 2 hr. For the agonist treatment experiment 1000 cells were seeded as described, then treated +/-100 µM IBMX before adding 10 µM forskolin following a time course of 2 hr. Intracellular cAMP was measured as described above. For shRNA infected cells, 48 hr post infection cells were treated with 100 µM IBMX and assayed for intracellular cAMP by radioimmunoassay as described (*Ho et al., 2015*). Where indicated cells were treated with 10 µg/ml anti-PTHrP neutralizing antibody at 37°C for 5 hr (*Onuma et al., 2004*). To compare cAMP levels between different cells equal number of cells were seeded.

## siRNA knockdown

Cells were transfected 24 hr after seeding with Dharmacon On-Target Plus siRNA pools (GE Healthcare Life Sciences, 20 nM final concentration) complexed with DharmaFECT3 (GE Healthcare Life Sciences) in Opti MEM reduced serum media (Life Technologies). Cells transfected with a non-targeting On-Target Plus control siRNA pool or mock transfected cells served as controls. All assays were carried out in culture medium without antibiotics. Cells were transfected for 72 hr with siRNA pool directed against *Creb1* along with the non-targeting control siRNA. After 72 hr incubation with siRNA, the control and the *Creb1* knockdown cells were treated with 10 µM of forskolin to assess *Creb1* target gene induction.

### ON-TARGETplus smart pool siRNA-*Creb1*

Target sequence 1:J-040959-12; Target sequence: UUUGUUAACUUCCGAGAAA
Target sequence 2:J-040959-11; Target sequence: GCUAUUGGCCUCCGGAAA
Target sequence 3:J-040959-10; Target sequence: GCUGAGUAUUAUAGCGUAU
Target sequence 4:J-040959-09; Target sequence: GAUAAGAGUAAGUCGAGA

## Plasmids and constructs

Two independent lentiviral shRNA plasmids targeting *Creb1* (A= 3'UTR; TRCN0000096629; B= CDS; TRCN0000096658) and *Pthlh* (A= 3'UTR; TRCN0000179093; B= CDS; TRCN0000180583) were purchased from Sigma-Aldrich in the pLKO.1-puro vector. The pLKO.1-puro sh-Luciferase plasmid (Cat No. SHC007) was used as the control for the Sigma shRNA constructs. Lentiviral packaging vector psPax2 (plasmid #12260) was obtained from Addgene (Cambridge, MA, USA), the pCMV-Eco Envelope (Cat No. RV112) vector was purchased from Cell Biolabs (San Diego, CA, USA). Three independent cell lines from each group were infected with shRNA against *Creb1*, *Pthlh* or luciferase control. After 48 hrs cells were selected with 2µg/ml puromycin.

### shRNA sequences

*Pthlh* 3' UTR (TRCN0000179093):
CCGGCCAATTATTCCTGTCACTGTTCTCGAGAACAGTGACAGGAATAATTGGTTTTTTG
*Pthlh* CDS (TRCN0000180583):
CCGGGATACCTAACTCAGGAAACCACTCGAGTGGTTTCCTGAGTTAGGTATCTTTTTG
*Creb1* 3' UTR (TRCN0000096629):
CCGGGCCTGAAAGCAACTACAGAATCTCGAGATTCTGTAGTTGCTTTCAGGCTTTTTG
*Creb1* CDS (TRCN0000096631):
CCGGCAGCAGCTCATGCAACATCATCTCGAGATGATGTTGCATGAGCTGCTGTTTTTG

*Luciferase* CDS (TRCN0000072259):
CCGGCGCTGAGTACTTCGAAATGTCCTCGAGGACATTTCGAAGTACTCAGCGTTTTT

## Alizarin staining for mineralization

Cells were washed 3 times with PBS and fixed with 70% EtOH. Cells were washed 3 times with water then stained with 0.5% alizarin (w/v) in water for 30 min at room temperature. This was followed by 3 washes with water and a 15 min wash in 1 ml of PBS while shaking. PBS was removed and plates were allowed to dry. Well images were then taken using an EPSON perfection V700 photo scanner. Alizarin dye was eluted overnight in 1 ml of 10% CTP (1-Hexadecylpyridinium chloride, w/v) in PBS with shaking. To construct a standard curve, a 1 mM solution of alizarin was dissolved in PBS aided by sonication, and 2-fold serial dilutions were made beginning from 400 µM. 200 µl of each serial dilution and 200 µl of eluted dye from each well were added separately to duplicate wells in a 96 well plate. The absorbance was read at OD562nm using a Polarstar optima+ microplate reader.

## Flow cytometry

OS cell lines (less than passage 5 from establishment) were prepared by trypsinization. Antibodies against murine CD45, Mac1, Gr1, F4/80, B220, IgM, CD2, CD3, CD4, CD8, Ter119, Sca1, CD51, PDGFRα (CD140a), CD31, either biotinylated or conjugated with eF450, PE, PerCP-Cy5.5, or APC were obtained from eBioscience (San Diego, CA) or Pharmingen. Biotinylated antibodies were detected with Streptavidin-Qdot605 (Invitrogen). Flow cytometry was performed on an LSRII Fortessa (BD Bioscience) interfaced with CellQuest software, data was analyzed on FlowJo (TreeStar).

## Annexin V staining

OS cells (3 independently derived fibroblastic and osteoblastic OS cultures) and primary osteoblasts were infected with 2 independent shRNA against *Creb1* and *Pthlh*. shRNA directed against luciferase (sh-Luc) was used as the control. Cells were infected for 48–72 hr prior to harvesting by trypsinization. Cells were washed then stained in 1x Annexin Binding buffer (eBioscience) diluted 1:20 with Annexin V-APC (1 mg/ml) (eBioscience) and 7-Aminoactinomycin D (7AAD) (100 µg/ml) (Life Technologies) for 15 min. Following the addition of 4 volumes of 1x Annexin Binding buffer, apoptotic cells were detected and quantified using FACS (LSRFortessa). Live cells (Annexin V negative, 7AAD low) and cells in early and late stages of apoptosis (Annexin V positive, 7AAD low/high) were quantified.

## PTHrP overexpression

Murine HA tagged Pthlh cDNA was generated by gene synthesis (Integrated DNA Technologies) and cloned into the retroviral MSCV-IRES-Zeocin. Constructs were sequence verified. Retrovirus was generated by transient transfection of 293T cells (purchased from ATCC, no authentication performed, mycoplasma negative as tested by PCR based assay by the Victorian Infectious Diseases Reference Laboratory) using the EcoPac envelope plasmid as previously described (*Singbrant et al., 2014*). Primary osteoblasts were infected by spin-infection with 8 µg/mL polybrene (Sigma). Levels of PTHrP were assayed by radioimmunoassay using UMR106.01 cells as previously described (*Partridge et al., 1983*) (derived by TJ Martin, no authentication performed, not mycoplasma tested).

## RNA extraction, cDNA synthesis and Quantitative realtime PCR (QPCR)

RNA was extracted using RNA extraction kits with on-column DNase digestion (Qiagen, Limburg, Netherlands; Bioline, London, UK) or TriSure reagent (Bioline). cDNA was synthesised from total RNA using a Tetro cDNA synthesis kit (Bioline) or AffinityScript cDNA synthesis kit (Agilent Technologies, Santa Clara, CA, USA). Gene expression was quantified on a Stratagene Mx3000P QPCR system (Agilent) using Brilliant II SYBR green QPCR master mix (Agilent) with primers specific to genes of interest (Primer sequences in *Table 2*). Gene expression between samples was normalized to *β2m* expression. Relative expression was quantified using the comparative CT method ($2^{-(Gene\ Ct\ -\ Normalizer\ Ct)}$). Samples were amplified in duplicate.

**Table 1.** Genes within the cAMP interactome that overlap with SNV mutations within human OS.

| cAMP int. | cAMP int. | cAMP int. | cAMP int. | Overlap(SNVs) |
|---|---|---|---|---|
| ABCC4 | CREB5 | MAP2K1 | ROCK1 | ADCY1 |
| ACOX1 | CREBBP | MAP2K2 | ROCK2 | ADRA1A |
| ACOX3 | DRD1 | MAPK1 | RRAS | ADRA2B |
| ADCY10 | DRD5 | MC2R | RRAS2 | AKAP1 |
| ADCY2 | EDNRA | MEK1 | SLC9A1 | AKAP3 |
| ADCY3 | EP300 | MEK2 | SOX9 | AKAP5 |
| ADCY4 | EPAC2 | MYL9 | SSTR1 | AKAP6 |
| ADCY5 | FFAR2 | NFAT2 | SSTR2 | ANXA1 |
| ADCY6 | FOS | NFATC | SUCNR1 | ATP1A1 |
| ADCY7 | FSH | NFATC1 | TIAM1 | ATP2B1 |
| ADCY8 | FSHR | NFKB1 | TNNI3 | CACNA1D |
| ADCY9 | FXYD1 | NFKBIA | TSHR | CACNA1F |
| ADCYAP1R1 | FXYD2 | NPY | VAV | CACNA1S |
| ADORA1 | GABBR | NPY1R | | CFTR |
| ADORA2A | GHRL | NR1C1 | | DRD2 |
| AF6 | GHSR | ORAI1 | | F2R |
| AKAP2 | GLI1 | OXTR | | GIPR |
| AKT | GLI3 | PACAPRI | | GNAI3 |
| AMH | GLP1R | PAK1 | | GRIA2 |
| ANPRA | GNAS | PKA | | GRIN2A |
| ARAP3 | GPR109 | PLCE | | GRIN2B |
| BAD | GPR119 | PLD1 | | GRIN3A |
| BDNF | GPR81 | PLN | | HCN4 |
| BRAF | GRIA1 | PPP1C | | PDE10A |
| CACNA1C | GRIA3 | PPP1R12A | | PDE2A |
| CALM | GRIA4 | PPP1R1B | | PDE4B |
| CAMK2 | GRIN1 | PTCH1 | | PDE4D |
| CAMK4 | GRIN2C | PTGER2 | | PDE6B |
| CHRM1 | GRIN2D | RAC1 | | PIK3C2B |
| CHRM2 | GRIN3B | RAC2 | | PIK3CG |
| c-Jun | HCN2 | RAC3 | | PIK3R4 |
| CNGA1 | HHIP | RAF1 | | PIK3R6 |
| CNGA2 | HTR1 | RAP1A | | PPP2R2B |
| CNGA3 | HTR4 | RAP1B | | PPP2R3A |
| CNGA4 | HTR6 | RAPGEF3 | | PRKCA |
| CNGB1 | JNK | RAPGEF4 | | PRKCB |
| CNGB3 | JUN | RELA | | PTGER3 |
| CREB1 | KAT3 | RHOA | | RYR2 |
| CREB3 | LIPE | VIPR2 | | SSTR5 |

## Western blotting

Protein lysates were prepared in RIPA buffer (50 mM Tris-HCl pH 7.4, 1% NP-40, 0.5% sodium deoxycholate, 0.1% SDS, 150 mM NaCl, 2 mM EDTA, 50 mM NaF). Protein (10–25 μg) was electrophoresed on 10% Bis Tris or 4–12% Bis-Tris gradient NuPAGE Novex protein gels (Life Technologies) and

**Table 2.** Oligonucleotide sequences used in RT-PCR.

| Gene | Forward primer | Reverse primer |
|---|---|---|
| Kcne4 | GTTATGTCCTTCTATGGCGTTTTC | ATCATAGGTAGCGGCTTCATAGC |
| Il6 | AACAAGAAAGACAAAGCCAGAGTC | CTCCAGCTTATCTGTTAGGAGAGC |
| Cxcl1 | TCATAGCCACACTCAAGAATGGT | TTTGGACAATTTTCTGAACCAAG |
| Dusp1 | TCACGCTTCTCGGAAGGATA | TGATGTCTGCCTTGTGGTTG |
| Nfil3 | GAGAAGAAAGACGCCATGTATTG | AGCTCAGCTTTTAAAGTGGCATT |
| Usp2 | CTGAAGCGCTATACAGAATCGTC | AAACCAAGTTTTTCCTTCTCCAG |
| Gem | TGGGAGAAGATACATATGAGCGTA | GAGTAGACGATCAGATAGGCATCC |
| Foxc2 | GCCAGAGAAGAAGATCACTCTGA | CACTTTCACGAAGCACTCATTG |
| Efnb2 | GGGGTCTAGAATTTCAGAAGAACA | ATCTTGTCCAACTTTCATGAGGAT |
| Btg2 | GCTGTATCCGTATCAACCACAAG | GATGCGGTAAGACACTTCATAGG |
| Ddit4 | TTTCAGTTGACCCTGGTGCT | GATGACTCTGAAGCCGGTACTTAG |
| Lif | ACCTTGAGAAAATCTACCGAGAAGT | AAAAATTTCTCCATTTTTGGCATA |
| Plaur | ACAGAGCACTGTATTGAAGTGGTG | GAAAGGTCTGGTTGCTATGGAA |
| Nrp1 | TACCCTCATTCTTACCATCCAAGT | CCACGTAGTCATACTTGCAGTCTC |
| Nfkbiz | TAAACATCAAGAATGAGTGCAACC | GTTGGTATTTCTGAGGTGGAGAGA |
| Ifngr1 | GTGGGGAGATCCTACATACGAA | CTTGCCAGAAAGATGAGATTCC |
| Rnf122 | GTCTTCATGCTTAGCCTCATCTTC | CAGGTCCCATAGAGCTGTAACTTC |
| Ugdh | CCTTCCTATTTATGAGCCTGGATT | CCATATGTTTTTGTTGGTGTGTTC |
| Osbpl9 | GTGTTAGCTACCTTGGGACATCAT | AGAACTCTGGGACTGTATTTGGAG |
| Ier3 | AATTTTCACCTTCGACCCTCTC | TTGGCAATGTTGGGTTCC |
| Cebpd | TCCTGCCATGTACGACGAC | TGTGGTTGCTGTTGAAGAGGT |
| Vegfa | GAAACCATGAACTTTCTGCTCTCT | ACTTGATCACTTCATGGGACTTCT |
| Sox9 | AGAAGGAGAGCGAGGAAGATAAGT | CTTGACGTGTGGCTTGTTCTT |
| Fos | GCTATCTCCTGAAGAGGAAGAGAAA | AACGCAGACTTCTCATCTTCAAGT |
| Nr4a3 | GGTGCAGAAAAATGCAAAATATG | CTGTCTGTACGCACAACTTCCTTA |
| Dusp1 | TCACGCTTCTCGGAAGGATA | TGATGTCTGCCTTGTGGTTG |
| Sik2 | ACCTTGAGAAAATCTACCGAGAAGT | AAAAATTTCTCCATTTTTGGCATA |
| Junb | CATCAACATGGAAGACCAGGA | GTTCTCAGCCTTGAGTGTCTTCA |
| Gtpbp10 | CCAAGTGCTAGGAGAACTCAATAAA | GCTATGACTTTTAGGTCAAGGTGAA |
| Adamts1 | GACCAGGAAGCATAAGGAAGAAG | CGAGAACAGGGTTAGAAGGTAATG |
| Bmp8a | CTGAGTTCCGGATCTACAAAGAAC | AGCGTCTGAAGATCCAAAAAGA |
| Lst1 | ACAACCAATGATTTCCTGCTAAAT | AGATGAACAGGATGATGACAAGC |
| Dlec1 | TCTAGACAGCAAGTTAATGCGAAA | ACAGCTAAACGTCAGCTTTGAAC |
| Tnfrsf12a | GCTGGTTTCTAGTTTCCTGGTCT | GTCTCCTCTATGGGGGTAGTAAACTT |
| Golga3 | AAAAAGAACTCCAAATCAAGCAAG | CCTCAGACACAACTGAAGTGCTAC |
| Tcf7 | TTTCTCCACTCTACGAACATTTCA | CCTGAGGTCAGAGAATAAAATCCA |
| Aqp3 | ATCAACTTGGCTTTTGGCTTC | GCATAGATGGGCAGCTTGAT |
| Hmg20b | CTTTGTAGTGGCTGTCAAGCAG | CATTTGGGAGAATCTTCTTTCTTTT |
| Tex264 | GTCTACTATGACAACCCCCATACG | GAAGGAGAATATCTTGAAGCCAAA |
| Creb1 | CAAGTCCAAACAGTTCAGATTTCA | TGGTGCATCAGAAGATAAGTCATT |
| Id1 | GGTGAACGTCCTGCTCTACG | AGACTCCGAGTTCAGCTCCA |
| Dsip1 | GGTGAACGTCCTGCTCTACG | AGACTCCGAGTTCAGCTCCA |
| Rgs2 | GTCCTCAAAAGCAAGGAAAATCTA | CATCAAACTGTACACCCTCTTCTG |
| Nr4a1 | CTCCTCCACGTCTTCTTCCTC | CAGGGACTGCCATAGTACTCAGA |

*Table 2 continued on next page*

*Table 2 continued*

| Gene | Forward primer | Reverse primer |
|---|---|---|
| *Areg* | CACAGGGGACTACGACTACTCAG | TCTTCCTTTTGGGTTTTTCTGTAG |
| *Nr4a2* | ACTGAAATTACTGCCACCACTTCT | TGTGCATCTGAATGTCTTCTACCT |
| *Pepck* | AGTGAGGAAGTTCGTGGAAGG | GCCAACAGTTGTCATATTTCTTCA |
| *Bnip3l* | GTCTCTAAGCATGAGGAAGAGTGG | AGAAGGTGTGCTCAGTCGTTTT |

transferred to PVDF membrane (Merck Millipore, Billerica, MA, USA). Membranes were blocked in 5% skim milk in TBST (20 mM Tris, 150 mM NaCl, 0.1% Tween-20) for 1 hr before incubation with primary antibodies diluted in 5% skim milk in TBST overnight a 4°C, or for 1 hr at room temperature in the case of pan-ACTIN. All antibodies (p53 (1C12) Cell Signaling 2524) (Phospho-CREB (Ser133) Cell Signaling Technologies, #9198), Anti-CREB1 ChIP grade (ab31387) were used at 1:2000, except pan actin (Ab-5, Thermo Scientific, Waltham, MA, USA) that was used at 1:3000. Anti-PTHrP antibody (R88) was generated in house against PTHrP(1–15). IgG was extracted from whole serum using Protein G Sepharose 4B fast Flow (GE Healthcare Life Sciences, Cat number 17-0618-01). Protocol for PTHrP western is the same as described for CREB1 except 3% BSA was used for blocking instead of skim milk. Following four washes of 10 min in TBST, membranes were incubated with secondary goat-anti-mouse or goat-anti-rabbit HRP conjugated antibodies (Thermo Scientific, Waltham, MA, USA) diluted 1:10,000 in 5% milk in TBST for 1 hr at room temperature. Following four 10 min

**Table 3.** Primers for promoter regions for ChIP and antibody conditions.

| Gene promoter | Forward primer | Reverse primer |
|---|---|---|
| *Nr4a2* | CTGCCAACATGCACCTAAAGT | CTTAAAATCAGCCCCAGTCGT |
| *Nr4a1* | TTCTGTTTCTAGGGACAGTGCAT | ACCCTACTCCAAGAGCTATCCTTT |
| *Cga* | CTCTTCATAAGCTGTCCTTGAGGT | GGTAAATTCTACCCAGTGATTGGA |
| *Areg* | TGATAACTAAGGAAACTGAGGTCCA | TTTGGAGAGGGAAAAATAAAATCA |
| *Dpp10* | AAGATCAGGGACTGTGGTACTGA | GGAATAGTGCATGTTTCCTTCTG |
| *Cebpd* | CACGGTTCACTAGTTCTGGTCTC | CTGGAGCGAAATGAAAATCTG |
| *Ifgnr1* | CTATGGTTTCCAGGAGCTTCAGT | AACTTCAGTTTGAACATGCACCT |
| *Rnf22* | CTATGGTTTCCAGGAGCTTCAGT | AACTTCAGTTTGAACATGCACCT |
| *Gem* | AAGCCCTTTTTGTACAAGTGTGA | GAGTGGGACAGTTTCTGTTTGAG |
| *Foxc2* | TTATCCATCACTGCATTCAACAG | AGTAGGAAAGAGCCTGGAGATTTT |
| *Fos* | GGTGCATACAGGAAGACATAAGC | GCAAAGTCCTGAAACAAAACAA |
| *Jun* | AGCAAAGATTAGCAAAGGGAAAG | CCAACTTTGAATCTGACAACTCC |
| *Sox9* | AGCAAAGATTAGCAAAGGGAAAG | CCAACTTTGAATCTGACAACTCC |
| *Vegfa 1* | GGGTGATGATAACAACAATTTGG | GAATATGGGCACAACAATTCAGT |
| *Vegfa 2* | ATTTGAGGGAGTGAAGAACCAAC | AGTCTGTGCTCTGGGATTTGATA |
| *Aqp3* | AGTCAAGGGTCATAGCTCCAGAT | TGGACCCAGAAGTGAGTTTCTAA |
| *Plaur* | CCTCAAAGGCTTTCTGTAGGAAT | AGGGGAAAAACAAGTTGAAAGAG |
| *Tnfrs12a* | GTTGTGTCTGCCCCTCAAGT | TTGCCCTATCTCTGGGTCTG |
| *Il6* | TCCTTTCCTGTCTGGAAGATACA | GGCAAAGAGATAAGGAAAAAGGA |

| Ab directed against | Cell number per ChIP | Amount | Origin of Ab |
|---|---|---|---|
| Creb1 | $6 \times 10^6$ | 2 μg | Abcam (ab31387) |
| Phospho-Creb1 | $6 \times 10^6$ | 2 μg | Cell Signaling (#9198) |
| Pol II (phospho-S2) | $2 \times 10^6$ | 2 μg | Abcam (ab103968) |

washes with TBST, membranes were exposed to ECL Prime (GE Healthcare Life Sciences, Piscataway, NJ, USA) and exposed to x-ray film to detect the expression levels of proteins.

## Chromatin immunoprecipitation

$5\times10^6$ cells for each subtype were seeded and allowed to proliferate for 24 hr. An additional counting plate for each subtype was used as cell count control. Primary osteoblasts and OS cells were treated with 10 µM forskolin or 0.1% v/v DMSO for a time course of 2 hr or used in an unstimulated state (PTHrP- or forskolin- free). Adherent cells were fixed with 1% formaldehyde-PBS for 30 min at room temperature. Cross-linking was quenched by incubating cells with 0.125 M glycine diluted in PBS for 10 min at room temperature. Following two washes with PBS, cells were scraped and snap frozen as pellets at -80°C until use. Cell pellets were diluted in sonication buffer (1% SDS, 10 mM EDTA, 50 mM Tris-HCl pH 8.1) with protease inhibitors (Roche, Burlington, NC, USA) and the DNA sheared to lengths between 200–800 bp using a UCD-200 Bioruptor (Diagenode, Denville, NJ, USA) on high at 4°C for a total shearing time of 15 min (90 min of 10 s on and 50 s off). Cell debris was cleared by centrifugation at 13,000 rpm for 10 min at 4°C and supernatants were diluted 10-fold in ChIP dilution buffer (0.01% SDS, 1.1% Triton X-100, 1.2 mM ETA, 16.7 mM Tris-HCl pH 8.1, 167 mM NaCl) with protease inhibitors. After removing 1% input for the total number of cells of each sample as an input control, samples were incubated with either 2 µg CREB1 antibody (Abcam: (ab31387), 2 µg of phospho-CREB1 antibody (Cell Signaling: 9198), 2 µg of phospho-Pol II antibody Abcam: (ab103968) and 2 µg of control rabbit IgG (Merck Millipore), or no antibody overnight at 4°C with rotation. Complexes were collected for 1 hr at 4°C with rotation with 60 µl of protein A sepharose beads (Invitrogen) that had been pre-blocked for 1 hr in 1 mg/ml BSA and 20 µg/ml of yeast tRNA (Sigma Aldrich, R5636). Beads were washed one time each with Low Salt buffer (0.1% SDS, 1% Triton X-100, 2 mM EDTA, 20 mM Tris-HCl pH 8.1, 150 mM NaCl), High Salt buffer (0.1% SDS, 1% Triton X-100, 2 mM EDTA, 20 mM Tris-HCl pH 8.1, 500 mM NaCl) and LiCl buffer (0.25 M LiCl, 1% NP40, 1% deoxycholate, 1 mM EDTA, 10 mM Tris-HCl pH 8.1), followed by two washes with TE buffer. Protein-DNA complexes were eluted from the beads (0.1 M NaHCO₃, 1% SDS) at RT for 30 min with two rounds of elution. Protein was digested by incubation with 50 µg/mL of proteinase K at 45°C for 1 hr. RNA was digested by incubation with 10 µg/mL RNaseA at 37°C for 30 min. DNA was purified by two extractions with phenol:chloroform:isoamyl alcohol and ethanol precipitation. CREB1, pCREB1 and Pol II bound and input samples were analysed by QPCR using primers that amplified CREB1 target gene promoters or negative control regions (Sequences in *Table 3* and for respective ChIP conditions refer to *Table 3*)

## Bioinformatics and data mining

### Somatic SNV mutations within human OS and enriched cAMP related functional pathways

Somatic SNV mutations derived from WGS of human osteosarcoma was downloaded (*Chen et al., 2014*). 1704 somatic SNV mutations comprising of insertions and deletions were subjected to pathway analysis. Analysis for functional pathways was performed using Cytoscape v3.1.1 (www.cytoscape.org) (*Shannon et al., 2003*; *Saito et al., 2012*). Highly enriched pathways within the cAMP signaling were selected based on FDR. cAMP interactome data containing 169 genes within the Kegg pathway were downloaded (entry number: map04024). The data set was overlapped using Biovenn and Venny (*BioinfoGP, CNB-CSIC* Key: citeulike: 6994833) with SNV mutations derived from WGS of human osteosarcoma. The overlap between the two databases was considered. Please refer to the *Table 1* for the gene sets and the overlaps.

### Enrichment of somatic SNV mutations within the cAMP interactome

cAMP interactome data containing 169 genes within the Kegg pathway were downloaded (entry number: map04024). Data from the indicated human tumor sets was obtained from the analysed data files and compared as total somatic SNV mutations (all) or predicted non silent SNV mutations as indicated. The tumor somatic SNV mutation data sets were overlapped using Biovenn and Venny (*BioinfoGP, CNB-CSIC* Key: citeulike: 6994833) with the cAMP interactome. Log p values are calculated using the hypergeometric distribution (phyper function in R). The human set size of 39,227 is derived using all symbols from HGNC.

## RNA-seq and data analysis

RNA-Sequencing (RNA-Seq) was conducted at the Ramaciotti Centre for Genomics (University of New South Wales, Australia) on the Illumina HiSeq 2000 with 100 bp paired-end reads. Reads were aligned to the mouse genome build mm9/NCBI37 using Casava 1.7 and Bowtie v0.12.2 mapping software, normalized using Voom linear modeling (*Law et al., 2014*) and transcript abundance measured as reads per kilobase of exon per million mapped reads (RPKM) (*Chepelev et al., 2009*). The datasets are deposited in GEO (accession number GSE58916).

### Statistical analysis

Data were presented as mean ± SEM. Statistical comparisons were performed in Prism 6.0 unless otherwise indicated. Parametric Student's *t*-test, area under the curve or 2-way ANOVA with multiple comparison test were used for comparisons with $p < 0.05$ considered as significant; Analysis of the enrichment for somatic SNV mutations within the cAMP interactome in each of the indicated tumor type was defined using hypergeometric distribution test (phyper function in R). *P* values as indicated in the Figure legend.

## Acknowledgements

We thank the SVH BioResources Centre for animal care; S Paton for technical assistance; L Purton and J Heierhorst for comments and discussion; SVI Flow Cytometry Facility (M Thomson) for assistance with FACS analysis; the neutralizing monoclonal antibody against PTHrP was kindly provided by Chugai Pharmaceutical Co., Gotemba, Japan. This work was supported by grants: NHMRC Project Grant APP1084230 (to CW, MW, TJM); Australian Sarcoma Study Group Johanna Sewell Research Grant (to CW, MW); Cancer Council of Victoria APP1047593 (CW); Zig Inge Foundation (CW); NHMRC Career Development Award APP559016 (CW); Victorian Cancer Agency Mid-Career Research Fellowship MCRF15015 (CW); Cancer Therapeutics CRC PhD Scholarship (AN); in part by the Victorian State Government Operational Infrastructure Support Program (to St. Vincent's Institute); CW was the Phillip Desbrow Senior Research Fellow of the Leukaemia Foundation.

## Additional information

### Funding

| Funder | Grant reference number | Author |
|---|---|---|
| National Health and Medical Research Council | APP1084230 | Mannu K Walia<br>T John Martin<br>Carl R Walkley |
| Australian Sarcoma Study Group | Johanna Sewell Research Grant | Mannu K Walia<br>Carl R Walkley |
| Cancer Council Victoria | APP1047593 | Carl R Walkley |
| Cancer Therapeutics CRC PhD Scholarship | | Alvin JM Ng |
| Department of Health, State Government of Victoria | Operational Infrastructure Support Scheme | T John Martin<br>Carl R Walkley |
| Leukaemia Foundation | Phillip Desbrow Senior Research Fellowship | Carl R Walkley |
| National Health and Medical Research Council | APP559016 | Carl R Walkley |
| Victorian Cancer Agency | Mid Career Research Fellowship MCRF15015 | Carl R Walkley |

The funders had no role in study design, data collection and interpretation, or the decision to submit the work for publication.

## Author contributions

MKW, Conceived study, Performed experiments, Analyzed and interpreted data, Wrote the manuscript; PMWH, ST, AJMN, Performed experiments, Analyzed and interpreted data, Drafting or revising the article; AG, Performed experiments, Analyzed and interpreted data, Acquisition of data, Analysis and interpretation of data, Drafting or revising the article; AMC, Performed experiments, Analyzed and interpreted data, Conception and design, Drafting or revising the article; ACWZ, Performed experiments, Analyzed and interpreted data, Provided samples and reagents, Drafting or revising the article, Contributed unpublished essential data or reagents; TJM, Conceived study, Performed experiments, Analyzed and interpreted data, Provided intellectual input and conceptual advice, Wrote the manuscript; CRW, Conceived study, Provided intellectual input and conceptual advice, Wrote the manuscript, Acquisition of data, Analysis and interpretation of data

## Author ORCIDs

Carl R Walkley, http://orcid.org/0000-0002-4784-9031

## Ethics

Human subjects: Primary human osteoblasts were isolated from bone marrow aspirates from the posterior iliac crest of de-identified healthy human adult donors with informed consent and consent to publish (IMVS/SA Pathology normal bone marrow donor program RAH#940911a, Adelaide, South Australia).

Animal experimentation: This study was performed in strict accordance with the recommendations in the Guide for the Care and Use of Laboratory Animals of the National Health and Medical Research Council, Australia and the Bureau of Animal Welfare, Victorian Government. All of the animals were handled according to approved institutional animal care and use committee (Animal Ethics Committee) protocols (#017/15) of the St. Vincent's Hospital Melbourne.

# Additional files

## Major datasets

The following previously published datasets were used:

| Author(s) | Year | Dataset title | Dataset URL | Database, license, and accessibility information |
|---|---|---|---|---|
| Chen X, Bahrami A, Pappo A, Easton J, Dalton J, Hedlund E, Ellison D, Shurtleff S, Wu G, Wei L, Parker M, Rusch M, Nagahawatte P, Wu J, Mao S, Boggs K, Mulder H, Yergeau D, Lu C, Ding L, Edmonson M, Qu C, Wang J, Li Y, Navid F, Daw NC, Mardis ER, Wilson RK, Downing JR, Zhang J, Dyer MA, St. Jude Children's Research Hospital-Washington University Pediatric Cancer Genome Project | 2014 | Recurrent somatic structural variations contribute to tumorigenesis in pediatric osteosarcoma | https://www.ebi.ac.uk/ega/studies/EGAS00001000263 | Publicly available at the European Bioinformatics Institute (accession no. EGAS00001000263) |

| | | | | |
|---|---|---|---|---|
| Berger MF, Hodis E, Heffernan TP, Deribe YL, Lawrence MS, Protopopov A, Ivanova E, Watson IR, Nickerson E, Ghosh P, Zhang H, Zeid R, Ren X, Cibulskis K, Sivachenko AY, Wagle N, Sucker A, Sougnez C, Onofrio R, Ambrogio L, Auclair D, Fennell T, Carter SL, Drier Y, Stojanov P, Singer MA, Voet D, Jing R, Saksena G, Barretina J, Ramos AH, Pugh TJ, Stransky N, Parkin M, Winckler W, Mahan S, Ardlie K, Baldwin J, Wargo J, Schadendorf D, Meyerson M, Gabriel SB, Golub TR, Wagner SN, Lander ES, Getz G, Chin L, Garraway LA | 2012 | Melanoma genome sequencing reveals frequent PREX2 mutations | http://www.ncbi.nlm.nih.gov/projects/gap/cgi-bin/study.cgi?study_id=phs000452.v2.p1&phv=167386&phd=&pha=&pht=2591&phvf=&phdf=&phaf=&phtf=&dssp=1&consent=&temp=1 | Publicly available at dbGaP (accession no. phs000452.v1.p1) |
| Cazier JB, Rao SR, McLean CM, Walker AK, Wright BJ, Jaeger EE, Kartsonaki C, Marsden L, Yau C, Camps C, Kaisaki P, Oxford-Illumina WGS500 Consortium, Taylor J, Catto JW, Tomlinson IP, Kiltie AE, Hamdy FC | 2014 | Whole-genome sequencing of bladder cancers reveals somatic CDKN1A mutations and clinicopathological associations with mutation burden | https://ega-archive.org/search-results.php?query=egas00001000738 | Publicly available at European Genome-Phenome Archive (accession no. EGA S00001000738) |
| Morin RD, Mungall K, Pleasance E, Mungall AJ, Goya R, Huff RD, Scott DW, Ding J, Roth A, Chiu R, Corbett RD, Chan FC, Mendez-Lago M, Trinh DL, Bolger-Munro M, Taylor G, Hadj Khodabakhshi A, Ben-Neriah S, Pon J, Meissner B, Woolcock B, Farnoud N, Rogic S, Lim EL, Johnson NA, Shah S, Jones S, Steidl C, Holt R, Birol I, Moore R, Connors JM, Gascoyne RD, Marra MA | 2013 | Mutational and structural analysis of diffuse large B-cell lymphoma using whole-genome sequencing | http://www.ncbi.nlm.nih.gov/projects/gap/cgi-bin/dataset.cgi?study_id=phs000532.v5.p2&phv=173801&phd=&pha=&pht=2979&phvf=&phdf=&phaf=&phtf=&dssp=1&consent=&temp=1 | Publicly available at dbGAP (accession no. phs000532.v2.p1) |

| | | | | |
|---|---|---|---|---|
| Tirode F, Surdez D, Ma X, Parker M, Le Deley MC, Bahrami A, Zhang Z, Lapouble E, Grossetête-Lalami S, Rusch M, Reynaud S, Rio-Frio T, Hedlund E, Wu G, Chen X, Pierron G, Oberlin O, Zaidi S1, Lemmon G, Gupta P, Vadodaria B, Easton J, Gut M, Ding L, Mardis ER, Wilson RK, Shurtleff S, Laurence V, Michon J, Marec-Bérard P, Gut I, Downing J, Dyer M, Zhang J, Delattre O; St. Jude Children's Research Hospital–Washington University Pediatric Cancer Genome Project and the International Cancer Genome Consortium. | 2014 | Genomic landscape of Ewing sarcoma defines an aggressive subtype with co-association of STAG2 and TP53 mutations | https://ega-archive.org/search-results.php?query=EGAS00001000855 | Publicly available at the European Genome-Phenome Archive (accession no. EGA S00001000855) |
| Tirode F, Surdez D, Ma X, Parker M, Le Deley MC, Bahrami A, Zhang Z, Lapouble E, Grossetête-Lalami S, Rusch M, Reynaud S, Rio-Frio T, Hedlund E, Wu G, Chen X, Pierron G, Oberlin O, Zaidi S1, Lemmon G, Gupta P, Vadodaria B, Easton J, Gut M, Ding L, Mardis ER, Wilson RK, Shurtleff S, Laurence V, Michon J, Marec-Bérard P, Gut I, Downing J, Dyer M, Zhang J, Delattre O; St. Jude Children's Research Hospital–Washington University Pediatric Cancer Genome Project and the International Cancer Genome Consortium. | 2014 | Genomic landscape of Ewing sarcoma defines an aggressive subtype with co-association of STAG2 and TP53 mutations | https://ega-archive.org/search-results.php?query=EGAS00001000839 | Publicly available at the European Genome-Phenome Archive (accession no. EGA S00001000839) |

| | | | | |
|---|---|---|---|---|
| Ellis MJ, Ding L, Shen D, Luo J, Suman VJ, Wallis JW, Van Tine BA, Hoog J, Goiffon RJ, Goldstein TC, Ng S, Lin L, Crowder R, Snider J, Ballman K, Weber J, Chen K, Koboldt DC, Kandoth C, Schierding WS, McMichael JF, Miller CA, Lu C, Harris CC, McLellan MD, Wendl MC, DeSchryver K, Allred DC, Esserman L, Unzeitig G, Margenthaler J, Babiera GV, Marcom PK, Guenther JM, Leitch M, Hunt K, Olson J, Tao Y, Maher CA, Fulton LL, Fulton RS, Harrison M, Oberkfell B, Du F, Demeter R, Vickery TL, Elhammali A, Piwnica-Worms H, McDonald S, Watson M, Dooling DJ, Ota D, Chang LW, Bose R, Ley TJ, Piwnica-Worms D, Stuart JM, Wilson RK, Mardis ER | 2012 | Whole-genome analysis informs breast cancer response to aromatase inhibition | https://www.ncbi.nlm.nih.gov/geo/query/acc.cgi?acc=GSE29442 | Publicly available at Gene Expression Omnibus (accession no. GSE29442) |
| Ellis MJ, Ding L, Shen D, Luo J, Suman VJ, Wallis JW, Van Tine BA, Hoog J, Goiffon RJ, Goldstein TC, Ng S, Lin L, Crowder R, Snider J, Ballman K, Weber J, Chen K, Koboldt DC, Kandoth C, Schierding WS, McMichael JF, Miller CA, Lu C, Harris CC, McLellan MD, Wendl MC, DeSchryver K, Allred DC, Esserman L, Unzeitig G, Margenthaler J, Babiera GV, Marcom PK, Guenther JM, Leitch M, Hunt K, Olson J, Tao Y, Maher CA, Fulton LL, Fulton RS, Harrison M, Oberkfell B, Du F, Demeter R, Vickery TL, Elhammali A, Piwnica-Worms H, McDonald S, Watson M, Dooling DJ, Ota D, Chang LW, Bose R, Ley TJ, Piwnica-Worms D, Stuart JM, Wilson RK, Mardis ER | 2012 | Whole-genome analysis informs breast cancer response to aromatase inhibition | https://www.ncbi.nlm.nih.gov/geo/query/acc.cgi?acc=GSE35191 | Publicly available at the Gene Expression Omnibus (accession no. GSE35191) |

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
