## [Decision Letter]

Thank you for submitting your work entitled "Activation of PTHrP-cAMP-CREB1 signaling following p53 loss is essential for osteosarcoma initiation and maintenance" for consideration by *eLife*. Your article has been favorably reviewed by three peer reviewers, and the evaluation has been overseen by a Reviewing Editor and Sean Morrison as the Senior Editor. One of the three reviewers has agreed to reveal his identity: Lawrence Donehower.

The reviewers have discussed the reviews with one another and the Reviewing Editor has drafted this decision to help you prepare a revised submission.

Summary:

The reviewers agree that your results strongly support an important role for autocrine/paracrine PTHrP-cAMP-CREB signaling in osteosarcoma pathogenesis. As you point out in the Introduction, increased PTHrP signaling in osteosarcoma has been observed previously, although PTHrP antibody treatment has not been effective in animal models (Ho et al., 2015). Your identification of the pathway as a key driver of osteosarcoma and the therapeutic opportunities it opens are very exciting discoveries. Conceptually, it would have been even more exciting if you could have somehow identified the molecular link(s) between loss of p53 and activation of PTHrP-cAMP-CREB signaling, but the reviewers are prepared to wait for another paper on this issue. Nonetheless, if you happen to have data that clarify this link, such data would strengthen the paper.

Essential revisions:

1) Tamoxifen-induced p53 deletion clearly causes an increase in PTHrP expression and subsequently cAMP production. The important results shown in Figure 1 should be expanded upon, and a cAMP time course should be shown following tamoxifen treatment. This experiment is important because the authors claim that derepression of the cAMP-CREB1 pathway is an "early event" following loss of p53, but only show cAMP levels at one time point.

2) The RNA-seq data in Figure 3 (an analysis of an already-published data set) shows nearly identical CREB levels in all 3 human osteoblast samples, with quite divergent levels in the OS samples. This important result should be expanded upon in more detail: do CREB levels change over the course of human OB differentiation? Ideally, additional normal samples would be included to strengthen this important finding. Along these lines, in all experiments where 'osteoblasts' are compared to various OS samples, the stage of osteoblast differentiation should be identified. The authors show clearly that CREB is down-regulated during osteoblast differentiation.

3) In addition to increased PTHrP levels in p53-depleted osteoblasts, reduced levels of PDEs, AKAPs, and PPs are observed. Is PTHrP overexpression alone sufficient to transform osteoblasts?

4) For all the experiments in Figure 4, the degree of cAMP upregulation due to forskolin treatment is likely to be much greater than that observed from autocrine/paracrine PTHrP. Cyclic AMP levels should be measured and directly compared prior to drawing any conclusions from this figure.

5) In Figure 6, a panel of 45 previously-determined PTHrP-responsive genes is measured in osteoblasts and OS cells. While it does appear that the PTHrP-responsive genes are more likely to be upregulated in osteoblastic OS, a more global comparison of gene expression in these samples would be helpful.

6) In Figure 7, the conclusion that several of the GO categories enriched within OS-associated SNVs are "cAMP related" is somewhat problematic. Since cAMP is an extremely well-studied second messenger, it's not surprising that cAMP signaling has been associated with many of the GO terms identified. More rigorous methods are needed to justify the claim made. Are the GO terms more enriched in terms related to cAMP than cGMP signaling (for example)?

7) Similarly, the overlap between OS-associated SNVs and genes in the KEGG cAMP interactome is provocative, but it also appears that other malignancies (lymphoma, breast) have a significant overlap as well (albeit with higher p values). Is this due to a pathogenic role for cAMP signaling in these cancers? We do not expect new experiments to definitely address this issue, but would appreciate clarification of the implications of this observation for your model in the text.

---

## [Author Response]

Summary:

*The reviewers agree that your results strongly support an important role for autocrine/paracrine PTHrP/cAMP/CREB signaling in osteosarcoma pathogenesis. As you point out in the Introduction, increased PTHrP signaling in osteosarcoma has been observed previously, although PTHrP antibody treatment has not been effective in animal models (Ho et al., 2015). Your identification of the pathway as a key driver of osteosarcoma and the therapeutic opportunities it opens are very exciting discoveries. Conceptually, it would have been even more exciting if you could have somehow identified the molecular link(s) between loss of p53 and activation of PTHrP/cAMP/CREB signaling, but the reviewers are prepared to wait for another paper on this issue. Nonetheless, if you happen to have data that clarify this link, such data would strengthen the paper.* We thank the editor and reviewers for their positive comments and for the opportunity to submit a revised version of our manuscript. We have incomplete preliminary data demonstrating that activation of the p53 response blunts the cAMP/CREB signalling response and would therefore only be in a position to speculate rather than provide a more definitive interpretation of the nature of the interaction. We are actively working on this and have initiated ChIP-seq experiments to resolve this at genome-wide resolution. We hope that you will find the revised manuscript with the additional data and text modifications as requested suitable for publication in *eLife*.

Essential revisions:

*1) Tamoxifen-induced p53 deletion clearly causes an increase in PTHrP expression and subsequently cAMP production. The important results shown in Figure 1 should be expanded upon, and a cAMP time course should be shown following tamoxifen treatment. This experiment is important because the authors claim that derepression of the cAMP-CREB1 pathway is an "early event" following loss of p53, but only show cAMP levels at one time point.*

We have included data from a detailed time course using primary osteoblasts isolated from *R26*-CreER^T2ki/+^*p53^+/+^*(WT) and *R26*-CreER^T2ki/+^*p53^fl/fl^*(KO) (new data – Figure 1). in vitrotamoxifen treatment was used to induce deletion of p53. Over 21 days culture, a loss of expression of p53 led to an acute rise in cAMP levels as compared to both wild-type controls and the non-tamoxifen treated isogenic cultures. *R26*-CreER^T2ki/+^*p53^+/+^*(WT) did not show any changes in cAMP levels compared to non-tamoxifen treated cells at the end of the time course. These data are consistent with the qPCR results in Figure 1 and indicate that the derepression of cAMP levels, and pathway activation as indicated by CREB1 transcriptional targets, is an "early event" following loss of p53. Note the new cAMP data replaces the original data and has been normalized to protein content, performed with a cAMP radioimmunoassay.

*2) The RNA-seq data in Figure 3 (an analysis of an already-published data set) shows nearly identical CREB levels in all 3 human osteoblast samples, with quite divergent levels in the OS samples. This important result should be expanded upon in more detail: do CREB levels change over the course of human OB differentiation? Ideally, additional normal samples would be included to strengthen this important finding. Along these lines, in all experiments where 'osteoblasts' are compared to various OS samples, the stage of osteoblast differentiation should be identified. The authors show clearly that CREB is down-regulated during osteoblast differentiation.*

The data used to assess *CREB1* levels in Figure 3 was obtained from a publically available, previously published data set used in Moriarty et al. (Nat Genet. 2015 Jun;47(6):615-24) and this data set only contained 3 normal osteoblast samples. Whilst is would not be possible to add additional control osteoblast samples directly to this data set, we have now assessed the status of CREB1 expression (both transcript and protein) in 5 independent human primary osteoblasts samples obtained from normal human bone. Cells from independent donors were subjected to osteogenic differentiation conditions for 28 days. By transcript expression *CREB1* reduces concomitant with the induction of osteoblast differentiation (as assessed by induction of transcripts associated with the transition to mature osteoblasts/osteocytes). This result in primary human osteoblast is highly comparable to the data we obtained with the primary murine long bone osteoblasts and supports that interpretation that in normal osteoblastic cells CREB1 levels reduce as the cells undergo differentiation. The new data is in Figure 3—figure supplement 2.

The populations we refer to when we describe “primary osteoblastic cells” are cells isolated from crushed, collagenase digested murine lone bones (tibia/femur). These cells are isolated and plated onto tissue culture dishes, allowed to expand for 5-7 days and then used for experiments. By flow cytometry, these cells are negative for haematopoietic markers (CD45, CD11b, F4/80), negative for the endothelial cell surface marker CD31 and co-express CD51 and Sca-1. The majority of the cells have a cell surface phenotype consistent with pre-osteoblasts (lin-CD45-CD31-CD51+Sca1+) when the cultures are initiated and when induced to differentiate acquire a mature osteoblast/osteocyte gene expression profile. Whilst still relative heterogeneous (at least by haematopoietic population standards) these cells are a considerably more pure and comparable population to the osteosarcoma derived cells than any other alternative that doesn’t involve FACS sorting. We have included the following text:

“As a control population (referred to herein as “primary osteoblasts”), we isolated osteoblastic cells from the collagenase digested long bones of wild-type C57BL/6 mice. […] The majority of the cells have a cell surface phenotype consistent with pre-osteoblasts (lin-CD45-CD31-CD51+Sca1+) when the cultures are initiated and when induced to differentiate acquire a mature osteoblast/osteocyte gene expression profile.”

*3) In addition to increased PTHrP levels in p53-depleted osteoblasts, reduced levels of PDEs, AKAPs, and PPs are observed. Is PTHrP overexpression alone sufficient to transform osteoblasts?*

We thank the reviewers for this comment as it has prompted us to explore the direct role of PTHrP in primary osteoblasts. To access if PTHrP overexpression alone was sufficient to immortalize normal cells, we retro-virally expressed PTHrP in normal osteoblasts. To confirm the we had in fact overexpressed PTHrP, we performed a bioassay for PTHrP in the media of the infected cells and empty vector infected control osteoblasts (note that this assay is performed by collecting the tissue culture media from the infected cells and assaying cAMP production on an independent cell line, UMR106.01, that constitutively express PTHR1). This analysis confirmed overexpression of PTHrP.

When we assessed the primary long bone osteoblasts overexpressing PTHrP we were surprised to find that they were poorly proliferative with a large increase in the proportions of dead cells (as assessed by AnnexinV/7AAD staining). This result was reproducible and occurred in independent cultures. We therefore conclude that increased PTHrP alone is not able to immortalise osteoblastic cells. New data in Figure 2—figure supplement 2.

*4) For all the experiments in Figure 4, the degree of cAMP upregulation due to forskolin treatment is likely to be much greater than that observed from autocrine/paracrine PTHrP. Cyclic AMP levels should be measured and directly compared prior to drawing any conclusions from this figure.*

We agree that forskolin is a supra-physiological inducer of cAMP. We have now included direct measurement of intracellular cAMP. The levels induced by forskolin are significantly higher, likely the maximal stimulation of this pathway in these cells. We have modified the text (subsection “Constitutively active cAMP differentially impacts primary osteoblasts and p53-deficient OS”, and new data Figure 4—figure supplement 2) to reflect the measurements but believe that these experiments are still worthwhile and the biology reflects an important observation of the effects of elevated cAMP in osteoblasts compared to osteosarcoma cells. As PTHR1 levels are different between the wild-type osteoblasts and also between the OS subtypes (osteoblastic OS express more PTHR1 than fibroblastic OS) we used forskolin to allow a more meaningful comparison of the effects of elevating cAMP.

*5) In Figure 6, a panel of 45 previously-determined PTHrP-responsive genes is measured in osteoblasts and OS cells. While it does appear that the PTHrP-responsive genes are more likely to be upregulated in osteoblastic OS, a more global comparison of gene expression in these samples would be helpful.*

We thank the reviewers for this suggestion. We have previously published an extensive characterisation of and comparison between the fibroblastic and osteoblastic OS models (see Mutsears et al., Bone 2013 Jul;55(1):166-78). This work included a global comparison of gene expression between these models and pathway analysis (GSEA). This comparison was limited to the two tumor subtypes. We do not presently have RNA-seq or microarrays of the primary osteoblasts as we have utilised in the present studies to allow a comparison across normal osteoblastic cells, fibroblastic and osteoblastic OS. Profiling of these normal populations is planned but has not been completed to date.

*6) In Figure 7, the conclusion that several of the GO categories enriched within OS-associated SNVs are "cAMP related" is somewhat problematic. Since cAMP is an extremely well-studied second messenger, it's not surprising that cAMP signaling has been associated with many of the GO terms identified. More rigorous methods are needed to justify the claim made. Are the GO terms more enriched in terms related to cAMP than cGMP signaling (for example)?*

We thank the reviewers for this comment as it is an important consideration. We have reassessed the enrichment of somatic SNVs with the tumors datasets within the cAMP, cGMP and a data set that is exclusive to each (i.e. with the genes common to both pathways removed and the respective sets reassessed). This analysis confirms our original finding that there is a significant and profound enrichment of somatic SNVs within the cAMP pathway in OS. We see some enrichment of cGMP pathways components, but to a much lesser extend that the cAMP pathway components. This is not unique to this tumor type, but is most statistically enriched within OS of the tumor types that we have assessed. We have included this as a new figure supplement (Figure 7—figure supplement 1).

7) Similarly, the overlap between OS-associated SNVs and genes in the KEGG cAMP interactome is provocative, but it also appears that other malignancies (lymphoma, breast) have a significant overlap as well (albeit with higher p values). Is this due to a pathogenic role for cAMP signaling in these cancers? We do not expect new experiments to definitely address this issue, but would appreciate clarification of the implications of this observation for your model in the text.

We do not mean to represent the activation of cAMP related pathways, and PKA in particular, as an OS unique event. It is well described in the contexts of other tumours, breast and haematological malignancies amongst other tumors, that this pathway is activated. We have amended the text, references and Discussion to reflect this. cAMP is regulated by the opposing effects of adenylyl cyclase and phosphodiesterases (PDEs). Both these components are regulated by a multitude of pathways like calcium signaling through calmodulin, G-proteins, inositol lipids and receptor tyrosine kinases. As many of these pathways are important for malignancies such as breast cancer and lymphoma, we would expect a significant enrichment of other malignancies within the cAMP component. What is notable is that OS is the highly enriched and the biological assessment of this pathway supports this.